# CROSS-TOKENIZER LIKELIHOOD SCORING ALGORITHMS FOR LANGUAGE MODEL DISTILLATION

**Buu Phan**[1]    **Ashish Khisti**[1]    **Karen Ullrich**[2]
[1]University of Toronto    [2]Meta AI
truong.phan@mail.utoronto.ca
akhisti@ece.utoronto.ca   karenu@meta.com

## ABSTRACT

Computing next-token likelihood ratios between two language models (LMs) is a standard task in training paradigms such as knowledge distillation. Since this requires both models to share the same probability space, it becomes challenging when the teacher and student LMs use different tokenizers, for instance, when edge-device deployment necessitates a smaller vocabulary size to lower memory overhead. This work addresses this vocabulary misalignment problem by uncovering an implicit recursive structure in the commonly deployed Byte-Pair Encoding (BPE) algorithm and utilizing it to create a probabilistic framework for *cross-tokenizer likelihood scoring*. Our method enables sequence likelihood evaluation for vocabularies different from the teacher model native tokenizer, addressing two specific scenarios: when the student vocabulary is a subset of the teacher vocabulary, and the general case where it is arbitrary. In the subset regime, our framework computes exact likelihoods and provides next-token probabilities for sequential sampling with only $\mathcal{O}(1)$ model evaluations per token. When used for distillation, this yields up to a $12\%$ reduction in memory footprint for the Qwen2.5-1.5B model while also improving baseline performance up to $4\%$ on the evaluated tasks. For the general case, we introduce a rigorous lossless procedure that leverages BPE recursive structure, complemented by a fast approximation that keeps large-vocabulary settings practical. Applied to GSM8K mathematical reasoning distillation, our method improves accuracy by over $2\%$ the current state of the art. Code: github.com/truongbuu/cross-tokenizer-scoring

## 1 INTRODUCTION

Many advanced training techniques for language models (LMs) rely on computing next-token likelihood ratios between two models. For example, reinforcement learning (Shao et al., 2024) and preference optimization (Zeng et al., 2024) use such ratios as bias-adjustment terms, ensuring that the learning signal received by the policy network is weighted properly. Similarly, in distillation (Hinton et al., 2015), the student model is trained to align with the teacher's next-token distribution to obtain richer learning signals and improved sample efficiency compared to supervised fine-tuning with human-annotated labels. However, computing these ratios is not always possible because it requires both LMs to share the same probability space, which can fail when the LMs employ different vocabulary—a misalignment issue that has recently drawn attention in LM distillation (Shin et al., 2025; Minixhofer et al., 2024; Zhang et al., 2024).

Consider the problem of LM distillation, due to the difference in output spaces, it becomes challenging to define what alternative objective could replace the divergence minimization objective between their next-token distributions. Importantly, we note that this misalignment also arises in the input space due to tokenization bias (Phan et al., 2025; Minixhofer et al., 2025), a phenomenon that determines which token cannot appear as a next-token. For example, suppose we use Llama3 as the teacher model, then given the input prompt `111+11=12`, tokenized as $[111, +, 11, =, 12]$ (comma-separated), the teacher model will never output `2` as the next token. This is because `122` is a single token in the Llama3 tokenizer, making the sequence $[12, 2]$ impossible in the tokenized text. Consequently, if the student model employs a byte-level tokenizer, this discrepancy creates a misleading training signal, degrading performance.

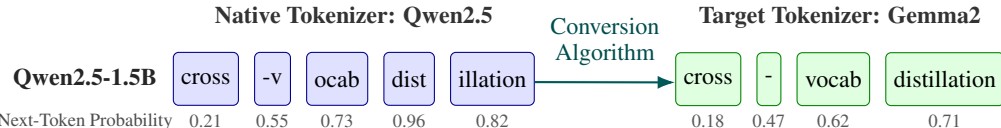

Figure 1: Cross-Tokenizer Scoring: given a language model trained on a fixed tokenizer, for example Qwen2.5, we aim to compute the probability that this model assigns to sequences encoded with a different tokenizer, e.g. Gemma2.

Current approaches to address this issue often introduce additional auxiliary loss functions, such as the Wasserstein distance in the logit space (Boizard et al., 2025), or incorporate extra components like sequence alignment (Minixhofer et al., 2025). Compared to scenarios where both models use the same tokenizer, these components introduce additional hyperparameters into the pipeline, deviating from the simplicity of Kullback–Leibler (KL) divergence minimization. In this work, we investigate the vocabulary misalignment problem through the lens of stochastic analysis and demonstrate that it is possible to re-align the teacher scoring model with the student vocabulary, see Figure 1, recovering the classic next-token divergence minimization/ likelihood ratio problem without introducing additional components. We refer to this process as *"cross-tokenizer scoring"* or *"conversion"* and provide a formal formulation in Section 4. Our analysis centers on the widely used Byte Pair Encoding (BPE) tokenization algorithm and demonstrates that if the student vocabulary is a subset of the one employed by the teacher, there is an efficient algorithm to convert the teacher model to score on the token sequences from the student vocabulary. This relates to the vocabulary trimming problem for small LMs (Cognetta et al., 2024; Ushio et al., 2023), which requires reducing the vocabulary size to satisfy resource constraints, i.e. minimizing the language modeling head memory footprint[1]. We then extend this result to the general cross-tokenizer scenario where the teacher and student vocabularies differ [2]. Our analysis yields a lossless but computationally intensive recursive algorithm to compute exact likelihoods in this setting, complemented by a more efficient beam search-based scheme that accurately approximates the ground-truth likelihood scores.

Overall, our contributions are as follows:

1. We develop a technical insight into the structure of the BPE algorithm and introduce a novel concept of *relative alphabets*. Leveraging this concept, we propose an $\mathcal{O}(1)$ next-token scoring algorithm for the case where the student vocabulary is a subset of the teacher's.

2. We provide an algorithm for conversion in the general case where teacher and student vocabularies differ, deriving a lossless recursive algorithm and a faster beam-search-based approximation.

3. We conduct extensive experiments to demonstrate the effectiveness of our methods in the cross-tokenizer distillation and vocabulary trimming task.

## 2 BACKGROUND

**Notations.** Let $\mathcal{A} = \{a_0, a_1, \ldots, a_{|\mathcal{A}|}\}$ denote the initial alphabet, where $a_0 = \text{EOS}$ is the end-of-string symbol unless stated otherwise. Following previous works (Phan et al., 2025; Vieira et al., 2025a), the EOS symbol marks the termination of a character sequence and is treated as a standalone token that does not appear as part of any other token. We distinguish between an infinite character sequence and a finite string $s$, which has length $|s|$. If a character sequence has EOS at index $x_i = \text{EOS}$, then all subsequent entries must also be EOS, i.e., $x_j = \text{EOS}$ for all $j > i$. Thus, effectively, the set of all sequences is countably infinite. In contrast, a finite string $s$ does not necessarily end with EOS. For a character sequence $\vec{x}$, we define its length $|\vec{x}|$ as the number of characters up to and including the first occurrence of EOS.

A vocabulary $\mathcal{V}$ is a list of tokens such that $\mathcal{A} \subseteq \mathcal{V}$, and we denote individual tokens as $t \in \mathcal{V}$. An encoding of a string $s$ is a sequence of tokens $\vec{e} = \text{encode}(s)$ of length $|\vec{e}|$, and the corresponding

---

[1]Recent work by Tao et al. (2024) on scaling laws for vocabulary size demonstrates a log-log relationship, suggesting that smaller language models benefit from smaller vocabularies.

[2]Both vocabularies are derived from the same base alphabet, i.e., UTF-8 bytes.

$\mathcal{V}_i$ is a *relative alphabet* of $\mathcal{V}_j$ for $i \le j$

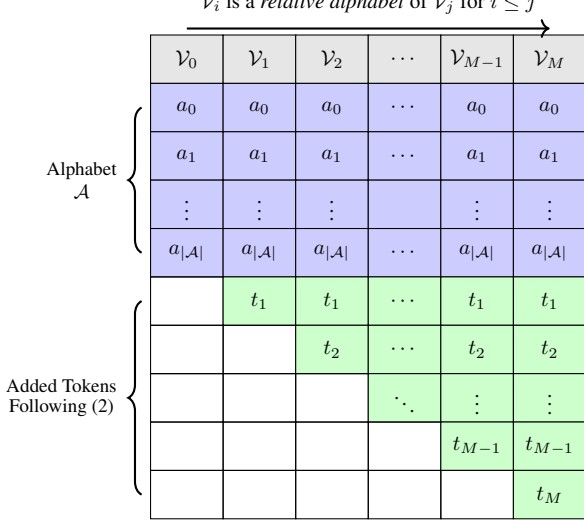

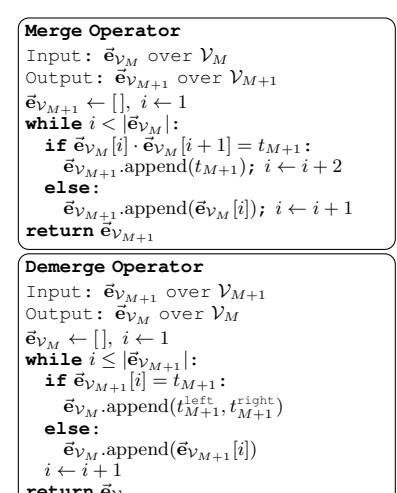

**Merge Operator**
```
Input: e⃗_{V_M} over V_M
Output: e⃗_{V_{M+1}} over V_{M+1}
e⃗_{V_{M+1}} ← [],  i ← 1
while i < |e⃗_{V_M}|:
    if e⃗_{V_M}[i] · e⃗_{V_M}[i+1] = t_{M+1}:
        e⃗_{V_{M+1}}.append(t_{M+1});  i ← i + 2
    else:
        e⃗_{V_{M+1}}.append(e⃗_{V_M}[i]);  i ← i + 1
return e⃗_{V_{M+1}}
```

**Demerge Operator**
```
Input: e⃗_{V_{M+1}} over V_{M+1}
Output: e⃗_{V_M} over V_M
e⃗_{V_M} ← [],  i ← 1
while i ≤ |e⃗_{V_{M+1}}|:
    if e⃗_{V_{M+1}}[i] = t_{M+1}:
        e⃗_{V_M}.append(t^left_{M+1}, t^right_{M+1})
    else:
        e⃗_{V_M}.append(e⃗_{V_{M+1}}[i])
    i ← i + 1
return e⃗_{V_M}
```

Figure 2: Progressive construction of BPE vocabulary and pseudocode for the merge and demerge operations. Newly added token at each step $j$ is constructed from two tokens within $\mathcal{V}_{j-1}$ and thus any prior vocabulary $\mathcal{V}_i$ for $i < j$ can be considered as *an alphabet* of $\mathcal{V}_j$.

decoding is the string $s = \text{decode}(\vec{e})$. Further details on how these mappings are defined are in Section 3. We use $\vec{e}[i:j]$ to denote the subsequence of tokens from index $i$ to $j$, inclusive. Also, $\vec{e} \cdot t$ denotes the encoding obtained by appending the token $t$ to the sequence $\vec{e}$. For an encoding $\vec{e}$, we use 1-based indexing. The slice $\vec{e}[i:j]$ includes elements from the $i$-th to the $j$-th position, totaling $j - i + 1$ elements. The notations $\vec{e}[:i]$ and $\vec{e}[i:]$ represent the subarrays from the start to the $i$-th element and from the $i$-th element to the end, respectively, both including $\vec{e}[i]$. The term $\vec{e}[-1], \vec{e}[-2], ...$ refers to the last, second last, etc., token in the encoding.

**Language Models.** We view a LM $P_{\mathcal{V}}$ over a vocabulary $\mathcal{V}$ as a stochastic process that generates tokens from $\mathcal{V}$ in an autoregressive manner. For an encoding $\vec{e}_{\mathcal{V}} \in \mathcal{V}^*$, $P_{\mathcal{V}}(\vec{e}_{\mathcal{V}})$ denotes the (joint) probability that this stochastic process generates a sequence with a prefix $\vec{e}$. Following previous works (Phan et al., 2025; Vieira et al., 2025b), a BPE encoding is valid (canonical) if and only if:

$$\vec{e} = \text{encode}(\text{decode}(\vec{e})), \tag{1}$$

otherwise it is invalid (non-canonical). Note that if $\vec{e}$ is invalid, then we have $P_{\mathcal{V}}(\vec{e}) = 0.0$. For the rest of this work, we assume that the given *LMs will always output valid encodings*, which is compatible with the model's behavior in practice (Vieira et al., 2025b).

## 3 SEQUENTIAL STRUCTURE IN BYTE-PAIR ENCODING

In this section, we build the foundational tools required for deriving the conversion algorithm. We begin by formalizing the structure of BPE encoding and decoding through the lens of subset vocabularies. We then introduce the key notion of the *relative alphabet*, which plays a central role in the conversion algorithms developed in Section 4.

### 3.1 SUBSET VOCABULARIES

We first review the structure of BPE vocabulary (Sennrich et al., 2015; Gage, 1994) and introduce the new concept of vocabulary subsets and sequential decoding through demerging.

**Definition 1.** Let $\mathcal{A}$ be the alphabet defined in Section 2. A *BPE vocabulary* $\mathcal{V}_M$ is an ordered list of $(|\mathcal{A}| + M)$ tokens consisting of the alphabet $\mathcal{A}$ and $M$ additional tokens constructed sequentially:

$$\mathcal{V}_M = \{a_0, a_1, \ldots, a_{|\mathcal{A}|},\ t_1, t_2, \ldots, t_M\},$$

where each merged token $t_i$ is defined recursively as

$$t_i = t_i^{\text{left}} \cdot t_i^{\text{right}}, \quad \text{with} \quad t_i^{\text{left}}, t_i^{\text{right}} \in \{a_1, \ldots, a_{|\mathcal{A}|}, t_1, \ldots, t_{i-1}\}. \tag{2}$$

We highlight that $t_i^{\text{left}}, t_i^{\text{right}} \neq \texttt{EOS}$ (i.e., $a_0$) for all $i$ and they must strictly follow the ordering requirement in (2). Also, Definition 1 does not concern the method used to obtain the vocabulary, i.e., which datasets are used to train the tokenizer.

**Definition 2** (BPE Vocabulary Subset). For any $M' \leq M$, the *truncated vocabulary* (or *BPE vocabulary subset*) is
$$\mathcal{V}_{M'} = \{a_0, a_1, \ldots, a_{|\mathcal{A}|},\ t_1, t_2, \ldots, t_{M'}\},$$
which contains the first $M'$ merged tokens of $\mathcal{V}_M$. We write $\mathcal{V}_{M'} \preceq \mathcal{V}_M$ to indicate that $\mathcal{V}_{M'}$ is a subset of $\mathcal{V}_M$ in the merge-order sense, and note that $\mathcal{V}_0 = \mathcal{A}$. Figure 2 (left) visualizes this definition.

**Notation Simplification.** We briefly note that the notations $\mathcal{V}_0, \mathcal{V}_1, \ldots \mathcal{V}_{M'}, \ldots, \mathcal{V}_M$ are reserved for vocabularies defined with construction in Definition 1 and 2 to avoid notation overwhelming. We also denote $P_0, P_1, ..P_M$ for the LM associated with these vocabularies. Otherwise, for any two vocabularies without this subset relation, we will use $\mathcal{V}_\alpha$ and $\mathcal{V}_\beta$.

**BPE Encoding and Decoding.** Let $\mathcal{V}_M$ be a BPE vocabulary constructed from an initial alphabet $\mathcal{A}$, with intermediate vocabularies $\mathcal{V}_i$ defined for $0 \leq i \leq M$. Given an input string $s$, the *encoding* of $s$ with respect to $\mathcal{V}_M$ is defined as:
$$\vec{\mathbf{e}}_{\mathcal{V}_M} = \text{encode}(s; \mathcal{V}_M), \quad \text{or simply} \quad \vec{\mathbf{e}}_{\mathcal{V}_M} = \text{encode}(s; M).$$
The encoding process begins from the character-level sequence $\vec{\mathbf{e}}_0 \in \mathcal{A}^*$ and applies a sequence of merge operations, shown in Figure 2 (right):
$$\vec{\mathbf{e}}_{\mathcal{V}_i} = \text{merge}(\vec{\mathbf{e}}_{\mathcal{V}_{i-1}}; \mathcal{V}_i) \triangleq \text{merge}_i(\vec{\mathbf{e}}_{\mathcal{V}_{i-1}}) \quad \text{for } i = 1, 2, \ldots, M, \tag{3}$$

where each merge applies the $i$-th merge rule in the construction of $\mathcal{V}_M$.

*Sequential Decoding.* The default decoding strategy simply maps each token to its surface alphabet form and concatenates the results. Instead, we reinterpret this process as a sequence of demerge operations. Specifically, we define:
$$s = \text{decode}(\vec{\mathbf{e}}_{\mathcal{V}_M}; \mathcal{V}_M) \quad \text{or simply} \quad s = \text{decode}(\vec{\mathbf{e}}_{\mathcal{V}_M}; M).$$
Starting from $\vec{\mathbf{e}}_{\mathcal{V}_M}$, we apply the inverse of the $i$-th merge rule using a demerge function associated with vocabulary $\mathcal{V}_i$, see Figure 2 (right):
$$\vec{\mathbf{e}}_{\mathcal{V}_{i-1}} = \text{demerge}(\vec{\mathbf{e}}_{\mathcal{V}_i}; \mathcal{V}_i) \triangleq \text{demerge}_i(\vec{\mathbf{e}}_{\mathcal{V}_i}) \quad \text{for } i = M, M-1, \ldots, 1. \tag{4}$$

The process terminates at the character-level string $\vec{\mathbf{e}}_{\mathcal{V}_0}$.

## 3.2 RELATIVE ALPHABET

Let $\mathcal{V}_{M'} \preceq \mathcal{V}_M$, and we are given an encoding $\vec{\mathbf{e}}_{\mathcal{V}_{M'}} = \text{encode}(s; M')$ of a string $s$, tokenized with $\mathcal{V}_{M'}$. We begin with the following question: how can we obtain the corresponding encoding of $s$ under the full vocabulary $\mathcal{V}_M$? A direct approach is to decode $\vec{\mathbf{e}}_{\mathcal{V}_{M'}}$ to recover the original string $s$, and then re-encode it using $\mathcal{V}_M$. This two-step procedure can be expressed as:
$$\text{(1) Decode: } s = \text{decode}(\vec{\mathbf{e}}_{\mathcal{V}_{M'}}; M') \quad \text{(2) Encode: } \vec{\mathbf{e}}_{\mathcal{V}_M} = \text{encode}(s; M). \tag{5}$$
This requires going through the intermediate representation, i.e. the string $s$, which is redundant. From Equation (3), we can compute directly $\vec{\mathbf{e}}_{\mathcal{V}_M}$ by continuing the merge operations from $i = M'+1$ to $M$. Conversely, to convert an encoding $\vec{\mathbf{e}}_{\mathcal{V}_M}$ under $\mathcal{V}_M$ to one under $\mathcal{V}_{M'}$, we can directly apply the demerge operations for $i = M, M-1, \ldots, M'+1$. Recall that the function $\text{encode}(.; M)$ and $\text{decode}(.; M)$ are also defined as a series of merge and demerge steps, we thus can view the subset vocabulary $\mathcal{V}_{M'}$ as an intermediate (or relative) alphabet of $\mathcal{V}_M$.

**Definition 3** (Relative Alphabet). If $\mathcal{V}_{M'} \preceq \mathcal{V}_M$, then $\mathcal{V}_{M'}$ is a *relative alphabet* with respect to $\mathcal{V}_M$. The *relative encoding* and *relative decoding* function that maps an encoding $\vec{\mathbf{e}}_{\mathcal{V}_{M'}}$ from $\mathcal{V}_{M'}$ to $\vec{\mathbf{e}}_{\mathcal{V}_M}$ from $\mathcal{V}_M$ and back are defined as:
$$\vec{\mathbf{e}}_{\mathcal{V}_M} = \text{encode}_{M' \to M}(\vec{\mathbf{e}}_{\mathcal{V}_{M'}}), \quad \vec{\mathbf{e}}_{\mathcal{V}_{M'}} = \text{decode}_{M \to M'}(\vec{\mathbf{e}}_{\mathcal{V}_M}), \tag{6}$$
where:
$$\text{encode}_{M' \to M}(\vec{\mathbf{e}}_{\mathcal{V}_{M'}}) = \text{merge}_M \circ \cdots \circ \text{merge}_{M'+1}(\vec{\mathbf{e}}_{\mathcal{V}_{M'}}), \tag{7}$$
$$\text{decode}_{M \to M'}(\vec{\mathbf{e}}_{\mathcal{V}_M}) = \text{demerge}_{M'+1} \circ \cdots \circ \text{demerge}_M(\vec{\mathbf{e}}_{\mathcal{V}_M}). \tag{8}$$

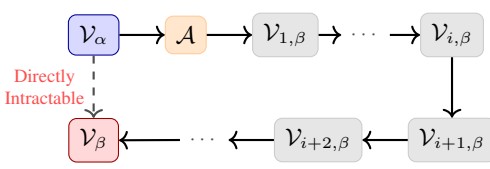

Figure 3: **Left:** Prior work (Phan et al., 2025) supports tractable conversion only from the full BPE vocabulary to the byte-level alphabet. **Right:** Our relative alphabet framework in Definition 3 enables conversion between any pair of BPE vocabularies by first mapping tokens to their byte-level representation, and then recursively applying reverse conversions between adjacent vocabularies (see Section 4.3 and Lemma 2), where $\mathcal{V}_{i,\beta}$ are the sub-vocabs containing the first $i$ merges of $\mathcal{V}_\beta$.

Following this definition, we now introduce the concept of *relative cover encodings*, which generalizes the notion of cover encodings from (Phan et al., 2025) to the case of relative alphabet.

**Definition 4** (Relative Cover Encodings). Let $\mathcal{V}_{M'} \preceq \mathcal{V}_M$, and let $\vec{\mathbf{e}}_{M'}$ be a valid encoding over the vocabulary $\mathcal{V}_{M'}$. We say that an encoding $\vec{\mathbf{e}}_{\mathcal{V}_M}$ over $\mathcal{V}_M$ is a *relative cover encoding* of $\vec{\mathbf{e}}_{\mathcal{V}_{M'}}$ if the following conditions hold:

1. There exists a sequence $\vec{\mathbf{x}}$ such that $\vec{\mathbf{e}}_{\mathcal{V}_M} = \mathrm{encode}(\vec{\mathbf{x}}; M)[: |\vec{\mathbf{e}}_{\mathcal{V}_M}|]$.

2. There exists an index $i \leq |\vec{\mathbf{e}}_{\mathcal{V}_{M'}}|$ such that:
$$\mathrm{decode}_{M \to M'}(\vec{\mathbf{e}}_{\mathcal{V}_M}[1:|\vec{\mathbf{e}}_{\mathcal{V}_M}| - 1]) = \vec{\mathbf{e}}_{\mathcal{V}_{M'}}[1:i-1],$$
$$\vec{\mathbf{e}}_{\mathcal{V}_{M'}}[i:] \in \mathrm{prefix}\left(\mathrm{decode}_{M \to M'}(\vec{\mathbf{e}}_{\mathcal{V}_M}[-1])\right),$$

   i.e. the last token of $\vec{\mathbf{e}}_{\mathcal{V}_M}$ starts from some location within $\vec{\mathbf{e}}_{\mathcal{V}_{M'}}$.

We denote by $\mathrm{cover}_{M' \to M}(\vec{\mathbf{e}}_{\mathcal{V}_{M'}})$ the set of all such relative cover encodings of $\vec{\mathbf{e}}_{\mathcal{V}_{M'}}$.

**Example 1.** Let $\mathcal{A} = \mathcal{V}_0 = \{\mathtt{a}, \mathtt{b}\}$, $\mathcal{V}_1 = \{\mathtt{a}, \mathtt{b}, \mathtt{a} \cdot \mathtt{b}\}$ and $\mathcal{V}_2 = \{\mathtt{a}, \mathtt{b}, \mathtt{a} \cdot \mathtt{b}, \mathtt{ab} \cdot \mathtt{a}\}$. Then we have $\mathrm{cover}_{1 \to 2}([\mathtt{a}, \mathtt{ab}]) = \{[\mathtt{a}, \mathtt{ab}], [\mathtt{a}, \mathtt{ab} \cdot \mathtt{a}]\}$. We do not include $\mathtt{EOS}$ in $\mathcal{A}$ for simplicity.

Finally, we note that one can efficiently search for all elements within this set with linear time complexity $\mathcal{O}(|\vec{\mathbf{e}}_{\mathcal{V}_{M'}}|)$ by leveraging the valid encoding condition (1). For details, see Appendix A.

## 4 CROSS-TOKENIZER LIKELIHOOD SCORING ALGORITHMS

This section provides a mathematical formalization of cross-tokenizer scoring (conversion) and presents the algorithms that enable this operation. The derivation of these algorithms makes essential use of the relative alphabet introduced in Section 3.2.

### 4.1 PROBLEM SETUP

**Cross-Tokenizer Scoring (Conversion).** Let $\mathcal{V}_\alpha$ and $\mathcal{V}_\beta$ be two BPE vocabularies built over the same base alphabet. We consider a language model $P_{\mathcal{V}_\alpha}$ trained with text tokenized under $\mathcal{V}_\alpha$. Given an encoding $\vec{\mathbf{e}}_{\mathcal{V}_\beta} \in \mathcal{V}_\beta^*$, our goal is to evaluate the probability, under $P_{\mathcal{V}_\alpha}$, of generating text whose $\mathcal{V}_\beta$-encoding begins with $\vec{\mathbf{e}}_{\mathcal{V}_\beta}$. Formally, let $\vec{\mathbf{e}}'_{\mathcal{V}_A} \sim P_{\mathcal{V}_\alpha}(\cdot)$ be the (random) token sequence obtained by autoregressively sampling from $P_{\mathcal{V}_\alpha}$ until the first occurrence of an $\mathtt{<EOS>}$ token, i.e. $\vec{\mathbf{e}}'_{\mathcal{V}_A}[-1] = \mathtt{<EOS>}$, and define the associated (random) byte sequence
$$\vec{\mathbf{x}} \triangleq \mathrm{decode}_{\mathcal{V}_\alpha}(\vec{\mathbf{e}}'_{\mathcal{V}_A}).$$

The *conversion probability* of the prefix $\vec{\mathbf{e}}_{\mathcal{V}_\beta}$ is then
$$P_{\mathcal{V}_\alpha}(\vec{\mathbf{e}}_{\mathcal{V}_\beta}) \triangleq \Pr\left[\mathrm{encode}_{\mathcal{V}_\beta}(\vec{\mathbf{x}}) \text{ begins with } \vec{\mathbf{e}}_{\mathcal{V}_\beta}\right].$$

That is, $P_{\mathcal{V}_\alpha}(\vec{\mathbf{e}}_{\mathcal{V}_\beta})$ denotes the probability that a sample from $P_{\mathcal{V}_\alpha}$, after decoding to its underlying byte sequence and re-encoding under $\mathcal{V}_\beta$, has $\vec{\mathbf{e}}_{\mathcal{V}_\beta}$ as a prefix.

In the conversion task, we would like to express $P_{\mathcal{V}_\alpha}(\vec{\mathbf{e}}_{\mathcal{V}_\beta})$ as a functions $f$ of $P_{\mathcal{V}_\alpha}(\vec{\mathbf{e}}_{i,\mathcal{V}_\alpha})$ for $i = 1, 2, ...$ where $\vec{\mathbf{e}}_{i,\mathcal{V}_\alpha} \in \mathcal{V}_\alpha^*$ are encodings within $\mathcal{V}_\alpha$, in particular:

$$P_{\mathcal{V}_\alpha}(\vec{\mathbf{e}}_{\mathcal{V}_\beta}) = f\Big( P_{\mathcal{V}_\alpha}(\vec{\mathbf{e}}_{1,\mathcal{V}_\alpha}),\ P_{\mathcal{V}_\alpha}(\vec{\mathbf{e}}_{2,\mathcal{V}_\alpha}),\ \dots \Big),$$

where each term $P_{\mathcal{V}_\alpha}(\vec{\mathbf{e}}_{i,\mathcal{V}_\alpha})$ is the standard autoregressive likelihood assigned by the model $P_{\mathcal{V}_\alpha}$ to the $\mathcal{V}_\alpha$-token sequence $\vec{\mathbf{e}}_{i,\mathcal{V}_\alpha}$, and can therefore be computed directly via a single forward pass of the model. Once this representation is established, next-token probabilities can be obtained directly via Bayes' rule. We illustrate the nontrivial nature of this task with the following example.

**Example 2.** Let $P_{\mathcal{V}_\alpha}$ operating over vocabulary $\mathcal{V}_\alpha = \{\mathsf{a}, \mathsf{b}\}$ and the target $\mathcal{V}_\beta = \{\mathsf{a}, \mathsf{b}, \mathsf{a}{\cdot}\mathsf{b}\}$. Suppose $\vec{\mathbf{e}}_{\mathcal{V}_\beta} = [\mathsf{b}, \mathsf{a}, \mathsf{b}]$, if we compute $P_{\mathcal{V}_\alpha}(\vec{\mathbf{e}}_{\mathcal{V}_\beta}) = P_{\mathcal{V}_\alpha}(\mathsf{b}) P_{\mathcal{V}_\alpha}(\mathsf{a}|\mathsf{b}) P_{\mathcal{V}_\alpha}(\mathsf{b}|\mathsf{a}, \mathsf{b})$ then we will end up with some positive probability if conditional probability is nonzero. This, however, is incorrect since the encoding $\vec{\mathbf{e}}_{\mathcal{V}_\beta} = [\mathsf{b}, \mathsf{a}, \mathsf{b}]$ never appears in the tokenized text distribution under $\mathcal{V}_\beta$, i.e. invalid encoding, and thus its probability is actually 0.0, i.e., see tokenization bias in (Phan et al., 2025).

**Algorithm Overview.** Let $\mathcal{V}_M$ be a BPE vocabulary and $\mathcal{V}_{M'} \preceq \mathcal{V}_M$ be one of its subsets. We will describe in details two types of conversions:

1. **Full-to-Subset Conversion:** Converts a language model $P_M$ trained on $\mathcal{V}_M$ to a statistically equivalent model $P_{M'}$ that operates on $\mathcal{V}_{M'}$.
2. **Subset-to-Full Conversion:** Converts a model $P_{M'}$ trained on $\mathcal{V}_{M'}$ to a statistically equivalent model $P_M$ that uses the full vocabulary $\mathcal{V}_M$.

For the general case with any pair of vocabularies $\mathcal{V}_\alpha$ and $\mathcal{V}_\beta$, we can:

1. Convert the model defined on $\mathcal{V}_\alpha$ to its byte-level equivalent on $\mathcal{V}_0$;
2. Convert the byte-level model on $\mathcal{V}_0$ to operate on the target vocabulary $\mathcal{V}_\beta$.

Combining the two scenarios, we can convert between any two BPE vocabularies $\mathcal{V}_\alpha$ and $\mathcal{V}_\beta$ that share the same base alphabet $\mathcal{A}$. We summarize this whole framework in Figure 3.

### 4.2 Full-to-Subset Conversion

Section 3.2 establishes that any subset vocabulary $\mathcal{V}_{M'}$ can be viewed an alphabet of $\mathcal{V}_M$. Thus, we can now leverage the previous result from (Phan et al., 2025), which converts tokenized LMs to byte-level (alphabet) LMs. In particular, we obtain the following conversion result that allows us to convert any model $P_{\mathcal{V}_M}$ over $\mathcal{V}_M$ to $P_{\mathcal{V}_{M'}}$ over $\mathcal{V}_{M'}$.

**Lemma 1.** Let $\mathcal{V}_{M'} \preceq \mathcal{V}_M$ with the based alphabet $\mathcal{A}$, and let $P(\cdot)$ be a language model over any arbitrary vocabulary $\mathcal{V}$ with $\mathcal{A} \subset \mathcal{V}$. Given a valid encoding $\vec{\mathbf{e}}_{\mathcal{V}_{M'}}$ over $\mathcal{V}_{M'}$, the probability that the LM $P(.)$ generates a string whose encoding under $\mathcal{V}_{M'}$ begins with $\vec{\mathbf{e}}_{\mathcal{V}_{M'}}$ is given by:

$$P(\vec{\mathbf{e}}_{\mathcal{V}_{M'}}) = \sum_{\vec{\mathbf{e}}_{\mathcal{V}_M} \in \mathcal{C}} P(\vec{\mathbf{e}}_{\mathcal{V}_M}), \tag{9}$$

where $\mathcal{C} \triangleq \mathrm{cover}_{M' \to M}(\vec{\mathbf{e}}_{\mathcal{V}_{M'}})$ is the set of relative cover encodings in Definition 4.

*Proof.* See Appendix B. ∎

From Lemma 1, LMs defined over the vocabulary $\mathcal{V}_M$ can be directly converted to operate on any sub-vocabulary $\mathcal{V}_{M'}$ by substituting $P(\cdot)$ with $P_M(\cdot)$. We conclude this full-to-subset conversion procedure with a discussion of the computational complexity of next-token sampling in this setting.

**Remark 1.** Lemma 1 can be seen as a generalization of the corresponding Lemma 1 in (Phan et al., 2025) by replacing $\mathcal{V}_{M'} = \mathcal{A}$. Furthermore, given that they share the same marginalization structure through the establishment of cover encodings, it is thus possible to perform autoregressive sampling with $\mathcal{O}(1)$ complexity in terms of model runs, allowing direct distillation for vocabulary trimming problems. For details, see Appendix A.

---

**Algorithm 1:** Recursive Vocabulary Conversion

---

**Input:** Encoding $\vec{e}_{\mathcal{V}_M}$ over vocabulary $\mathcal{V}_M$; language model $P$ defined over $\mathcal{V}_{M'} \preceq \mathcal{V}_M$
**Output:** Probability $P(\vec{e}_{\mathcal{V}_M})$ under $P$

1 **Function** ConvertProb($\vec{e}, M$):
2     **if** $M = M'$ **then**
3        **return** $P_{M'}(\vec{e})$;
4     Let $t_M = t_M^{\text{left}} \cdot t_M^{\text{right}}$ be the token added at step $M$;
5     $\vec{e}_{\text{dec}} \leftarrow \text{decode}_{M \rightarrow M-1}(\vec{e})$;
6     $\vec{e}_{\text{alt}} \leftarrow \vec{e}_{\text{dec}} \cdot t_M^{\text{right}}$;
7     **if** $\vec{e}[-1] \neq t_M^{\text{left}}$ **then**
8        **return** ConvertProb($\vec{e}_{\text{dec}}, M-1$);
9     **return** ConvertProb($\vec{e}_{\text{dec}}, M-1$) $-$ ConvertProb($\vec{e}_{\text{alt}}, M-1$);

---

**Example 3.** Following the setup in Example 1, then we can express $P_{\mathcal{V}_1}([\text{a}, \text{ab}])$, i.e. the probability of encoding within $\mathcal{V}_1$ as the sum of $P_{\mathcal{V}_2}([\text{a}, \text{ab}])$ and $P_{\mathcal{V}_2}([\text{a}, \text{aba}])$.

### 4.3 SUBSET-TO-FULL CONVERSION

For this scenario, it is unclear whether similar direct marginalization is possible. Consider the problem of converting $\mathcal{V}_0$ to $\mathcal{V}_M$, one idea is to perform the following marginalization:

$$P_0(\vec{e}_{\mathcal{V}_M}) = \sum_{|s|=\text{L}}^{\infty} P_0(s \cdot \text{EOS}) \mathbf{1}\{\vec{e}_{\mathcal{V}_M} = \text{encode}_M(s \cdot \text{EOS})[1 : |\vec{e}_{\mathcal{V}_M}|]\} \tag{10}$$

where $\text{L} = |\text{decode}_{M \rightarrow 0}(\vec{e}_{\mathcal{V}_M})|$. That is, we sum over the probability of the string $s$ ending with EOS, whose encoding under $\mathcal{V}_M$ begins with the desired prefix $\vec{e}_M$, ranging over arbitrarily long sequences. Carrying out this marginalization requires enumerating such strings without any *a priori* stopping bound, prompting a question of whether there is a finite-time procedure for this conversion problem. Indeed, by again leveraging the sequential decoding structure, we find that such a procedure exists and one can compute $P_0(\vec{e}_M)$ within finite enumeration. In particular, we adopt a recursive construction approach and iteratively convert encodings across incremental vocabularies where the conversion is tractable for each intermediate step. Specifically, let $\vec{e}_{\mathcal{V}_{M'+1}}$ be an encoding from $\mathcal{V}_{M'+1}$ and recall its construction:

$$\mathcal{V}_{M'+1} = \mathcal{V}_{M'}.\text{append}(t_{M'+1}) \quad \text{where: } t_{M'+1} = t_{M'+1}^{\text{left}} \cdot t_{M'+1}^{\text{right}} \tag{11}$$

Suppose the last token of $\vec{e}_{\mathcal{V}_{M'+1}}[-1] \neq t_{M'+1}^{\text{left}}$ then we have the cover encoding set size $|\mathcal{C}| = 1$ in Equation (9) since there is no token in $\mathcal{V}_{M'+1}$ can contain $\vec{e}_{\mathcal{V}_{M'+1}}[-1]$ except itself. Thus, let $\vec{e}_{\mathcal{V}_{M'}} = \text{decode}_{M'+1 \rightarrow M'}(\vec{e}_{\mathcal{V}_{M'+1}})$, then we have:

$$P_{M'}(\vec{e}_{\mathcal{V}_{M'+1}}) = P_{M'}(\vec{e}_{\mathcal{V}_{M'}}) \tag{12}$$

If $\vec{e}_{\mathcal{V}_{M'+1}}[-1] = t_{M'+1}^{\text{left}}$ then besides $t_{M'+1}^{\text{left}}$, we also have $t_{M'+1}$ containing $\vec{e}_{\mathcal{V}_{M'+1}}[-1]$, thus:

$$P_{M'}(\vec{e}_{\mathcal{V}_{M'+1}}) = P_{M'}(\vec{e}_{\mathcal{V}_{M'}}) - P_{M'}(\vec{e}_{\mathcal{V}_{M'}} \cdot t_{M'+1}^{\text{right}}), \tag{13}$$

and thus the recursion relation can be established, shown in Algorithm 1. The correctness proof is in Appendix C and it is guaranteed to terminate in finite time: each recursive call steps back to the immediately preceding vocabulary, progressing toward $\mathcal{V}_{M'}$. Nevertheless, its worst-case cost is $\mathcal{O}(\exp(M - M'))$ evaluations of $P_{M'}$, which is prohibitive. In practice, however, most leaves contribute negligibly because (i) well-trained LMs concentrate next-token probability on a small subset of candidates, and (ii) the vast majority of leaf encodings are semantically or syntactically implausible and therefore receive vanishing probability. We thus can prune the recursion by restricting to high-probability continuations and exploiting pre-tokenization patterns. A complete description is in Appendix C.1, with empirical results for approximation quality in Section 6.1.

**Example 4.** Following the setup in Example 1, say that we want to compute $P_{\mathcal{V}_2}([\texttt{a}, \texttt{ab}])$. Running the algorithm step by step, we have:

1. Recursion at $\mathcal{V}_2$: since in $\mathcal{V}_2$, the token $\texttt{ab}$ is a part of the final token $\texttt{ab} \cdot \texttt{a}$, we branch out:

$$P_{\mathcal{V}_2}([\texttt{a}, \texttt{ab}]) = P_{\mathcal{V}_1}([\texttt{a}, \texttt{ab}]) - P_{\mathcal{V}_1}([\texttt{a}, \texttt{ab}, \texttt{a}])$$

2. Recursion at $\mathcal{V}_1$: since in $\mathcal{V}_1$, the token $\texttt{a}$ in the second term of the above equation is a part of the final token $\texttt{a} \cdot \texttt{b}$, we branch out again while keeping the first term intact:

$$P_{\mathcal{V}_2}([\texttt{a}, \texttt{ab}]) = P_{\mathcal{V}_0}([\texttt{a}, \texttt{a}, \texttt{b}]) - [P_{\mathcal{V}_0}([\texttt{a}, \texttt{a}, \texttt{b}, \texttt{a}]) - P_{\mathcal{V}_0}([\texttt{a}, \texttt{a}, \texttt{b}, \texttt{a}, \texttt{b}])]$$
$$= P_{\mathcal{V}_0}([\texttt{a}, \texttt{a}, \texttt{b}]) - P_{\mathcal{V}_0}([\texttt{a}, \texttt{a}, \texttt{b}, \texttt{a}])$$

Intuitively, there are four possibilities for the next two characters of the string $\texttt{aab}$, namely $\texttt{aa}$, $\texttt{ab}$, $\texttt{ba}$ and $\texttt{bb}$. Consider the string $\texttt{aabaa}$, one can see that, regardless of any continuation after, the first two tokens under $\mathcal{V}_2$ is still $[\texttt{a},\texttt{aba}]$ due to the last character $\texttt{a}$ acts as a buffer guaranteeing the merge for token $\texttt{aba}$. For the string $\texttt{aabab}$, the last character $\texttt{b}$ prevents the merge of $\texttt{aba}$ and thus guarantee to form the desired encoding $[\texttt{a}, \texttt{ab}]$. Finally, the strings $\texttt{aabba}$ and $\texttt{aabbb}$ must have $[\texttt{a},\texttt{ab}]$ as the first two tokens under $\mathcal{V}_2$, thus consistent with final result.

## 5 RELATED WORK

**Cross-Tokenizer Sampling and Scoring.** There has been growing interest in sampling from language models under tokenizers different from the ones they were originally trained on. Phan et al. (2025) introduce the concept of *tokenization bias*, explaining why tokenized LMs can fail to produce correct outputs when prompts end with partial tokens. They further propose an algorithm that converts tokenized LMs into equivalent byte-level models, achieving constant-time complexity of $\mathcal{O}(1)$ (model runs) for next-token sampling. Similar studies (Vieira et al., 2025a; Hayase et al., 2025) also tackle this problem, using slightly different formulations and algorithmic techniques. Our work generalizes these prior approaches along two dimensions. First, whereas existing methods are restricted to conversions from a full vocabulary to the byte-level alphabet $\mathcal{A}$, our formulation in Section 4.2 applies to *any* subset vocabulary $\mathcal{V}_i \preceq \mathcal{V}_M$. Building on this formulation, we further develop algorithms for the cross-tokenizer conversion setting, see Section 4.3, enabling conversions between arbitrary pairs of BPE tokenizers. To the best of our knowledge, existing results for this setting have only been partially explored in Phan et al. (2024) for maximum-prefix encoding, which has been largely replaced by BPE. Second, at the level of analysis, we introduce new technical tools to characterize such conversions; e.g., the notion of *relative cover encodings* (Definition 4) generalizes the cover encoding of Phan et al. (2025) beyond the byte-level vocabulary $\mathcal{A}$.

**Cross-Tokenizer Distillation.** Cross-tokenizer distillation seeks to transfer knowledge across models built on distinct vocabularies. Prior work (Boizard et al., 2025; Cui et al., 2025; Feng et al., 2024) has approached this by applying optimal transport to align teacher and student logits, thereby providing vocabulary-agnostic supervision. A complementary line (Wan et al., 2024; Minixhofer et al., 2025; Chen et al., 2025) instead reduces distributional divergence by approximating marginalized probabilities through sequence alignment. Ultimately, these methods either impose strong assumptions on the logit representation or rely on heuristic approximations of sequence likelihoods. In contrast, our framework derives directly from first principles: we develop a probabilistic formulation that computes exact sequence likelihoods under arbitrary BPE vocabularies, enabling lossless and principled cross-tokenizer distillation.

## 6 EXPERIMENTS

### 6.1 CROSS-TOKENIZER DISTILLATION

**Approximation Quality.** We begin with validating the approximation scheme mentioned in Section 4.3. In particular, we convert the Qwen2.5–7B–Instruct model to its equivalent byte-level model using the Full-to-Subset procedure, where we can compute the next-byte probability over the whole alphabet with $\mathcal{O}(1)$ complexity. We use this byte-level probabilities from this converted model to re-compute the next-token probability of the original Qwen2.5–7B–Instruct model using

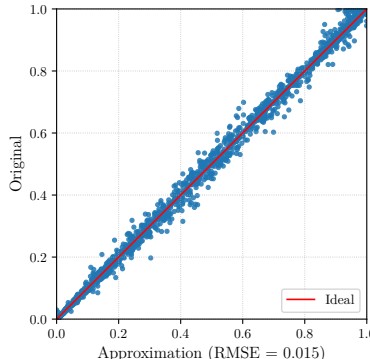

Figure 4: Empirical approximation of next-token probabilities.

Table 1: GSM8K distillation (Qwen2.5-Math-7B → Gemma2-2B).

| Method | Accuracy (5-shot) |
|---|---|
| Gemma2-2B-Instruct (Student) | 52.3 |
| Qwen2.5-Math-7B-Instruct (Teacher) | 88.4 |
| SFT | 47.9 |
| ULD (Boizard et al., 2025) | 47.1 |
| ULD + SFT (Boizard et al., 2025) | 48.2 |
| DSKD (Zhang et al., 2024) | 51.5 |
| DSKD + SFT (Zhang et al., 2024) | 52.8 |
| ALM | 53.2 |
| ALM + SFT | 53.5 |
| PKL (Ours) | 54.6 |
| SFT + PKL (Ours) | **55.6** |

the approximation process in Appendix C.1. Figure 4 shows the approximations closely match the token-level ground truths, confirming the correctness of the proposed method. This beam-search approximation amounts to evaluating approximately 6 beams, each with an average length of 10, and takes roughly 0.5 seconds per next-token evaluation, i.e. unlike the Full-to-Subset case, we only get the probability of one query token instead of the whole distribution over the subset vocabulary.

**Distillation Results.** We train and evaluate cross-tokenizer distillation on GSM8K using Qwen2.5-Math-7B-Instruct (teacher) and Gemma-2-2B-Instruct (student), chosen for their minimal vocabulary overlap. For each prompt, we score five tokens: the ground-truth, the student's top-1, and three additional candidate tokens obtained via beam search; see Appendix C.1. Training minimizes a partial Kullback–Leibler divergence (PKL) (weight $\omega$, see Equation (35) in Appendix D) plus a supervised-finetuning (SFT) term (weight $1 - \omega$); we use PKL instead of forward KL because only a small subset of (1 to 5) token probabilities is available. To minimize training-time computation, we performed teacher inference offline, generating soft labels for 7,500 training examples. We show results for $\omega \in \{0.0, 0.8, 1.0\}$ (pure SFT, validated mix, pure PKL) and compare against Approximate Likelihood Matching (ALM) (Minixhofer et al., 2024), the current state of the art for cross-tokenizer distillation. Models are trained end-to-end (without LoRA), using a batch size of 64 and a learning rate of $5 \times 10^{-6}$. Table 1 shows our method outperforms others and achieves the best results when combined with SFT. Notably, our approach strictly generalizes ALM: it recovers ALM in the limit of ALM chunk size 1 with (near-)perfect debiasing, while additionally leveraging probabilities of non–ground-truth tokens. See Appendix E for results on translation task.

## 6.2 VOCABULARY TRIMMING

We study the performance of our approach for the Full-to-Subset conversion in the vocabulary trimming problem, i.e. to decrease the memory footprint of its language modeling head. We evaluate our method's performance on domain-specific reasoning tasks, including arithmetic reasoning and coding, alongside with multiple general-purpose reasoning tasks in Appendix E.

**Setup.** We utilize the Qwen2.5-1.5B-Instruct model as the base for the student model, with an original vocabulary size of $151,643$, and truncate to the first $\{16, 32, 64\} \times 10^3$ merges. For both domain-specific and general-purpose reasoning tasks, we initiate a warm-up distillation phase for one epoch on the Alpaca dataset (Taori et al., 2023), which contains approximately 50,000 instruction-following examples, using the Qwen2.5-7B-Instruct model as the teacher model. The primary goal is to enable the student model to redistribute probability mass over its reduced vocabulary. We compare performance with $\omega = 0.0$ (pure SFT), $\omega = 1.0$ (pure distillation), and

| Vocab Size | LMH | LM Memory | Saved |
|---|---|---|---|
| 16k | 45 MB | 2.48GB | 13.5% |
| 32k | 91 MB | 2.53GB | 12.0% |
| 64k | 182 MB | 2.62GB | 9% |
| 151k | 445 MB | 2.88GB | 0% |

*LMH* = LM head memory.

Table 2: Memory impact of vocabulary trimming (bfloat16).

| Vocab | GSM8K (4-shot Acc) | | | | | HumanEval (0-shot) | | | | | MBPP (3-shot) | | | | |
|---|---|---|---|---|---|---|---|---|---|---|---|---|---|---|---|
| | SFT | FKL | ALM | ULD | DSDK | SFT | FKL | ALM | ULD | DSDK | SFT | FKL | ALM | ULD | DSDK |
| 16k | 53.2 | 56.5 | 57.0 | 51.2 | 52.1 | 43.2 | 42.6 | 34.7 | 28.6 | 30.5 | 30.8 | 40.8 | 34.8 | 32.2 | 33.3 |
| (w/ SFT) | | 59.7 | **60.4** | 53.6 | 54.3 | | **46.4** | 30.5 | 31.1 | 32.3 | | **41.4** | 31.4 | 33.1 | 34.2 |
| 32k | 54.0 | 58.6 | 57.1 | 53.5 | 54.4 | 46.9 | **47.6** | 32.3 | 30.5 | 40.2 | 38.8 | **41.8** | 35.6 | 33.5 | 34.6 |
| (w/ SFT) | | **63.0** | 61.1 | 57.1 | 59.3 | | 47.6 | 39.6 | 34.1 | 41.4 | | 40.6 | 33.4 | 32.2 | 36.0 |
| 64k | 54.5 | 61.8 | 56.2 | 55.5 | 57.3 | 48.2 | **51.8** | 31.7 | 32.3 | 42.6 | 39.9 | **44.6** | 39.0 | 34.4 | 37.0 |
| (w/ SFT) | | **62.8** | 61.5 | 58.4 | 60.2 | | 51.2 | 47.5 | 36.1 | 46.4 | | 43.0 | 36.4 | 35.2 | 38.7 |
| Full Vocab | 60.2 | 61.1 | 57.0 | 58.1 | - | 49.4 | 50.6 | 42.6 | 48.7 | - | 42.0 | 43.0 | 39.9 | 41.2 | - |
| (w/ SFT) | | **63.2** | 62.1 | - | - | | **52.4** | 48.2 | - | - | | **43.4** | 42.4 | - | - |

Table 3: Performance on GSM8K, HumanEval, and MBPP (pass@1). Columns report SFT, FKL (our method, with/without SFT), and the baselines ALM (with/without SFT), ULD and DSDK. In the Full Vocab row, the blue entry (ULD column) denotes the original model's performance, not a ULD score. Our method consistently improves accuracy under the vocabulary trimming setup.

set $\omega = 0.8$ for the combined loss after validation sweeps. Following warm-up, for mathematical reasoning, we distill the student for two epochs on the GSM8K training split (Cobbe et al., 2021), using Qwen2.5-Math-7B-Instruct as a domain-expert teacher. For the coding task, we distill the student models on the OPC dataset (Huang et al., 2024), employing the Qwen2.5-Coder-7B-Instruct model. To minimize on-the-fly computation, we perform teacher inference offline and store the top-$K$ probabilities for distillation, with $K = 20$.

**Results.** As shown in Table 3, our alignment method, consistent with the cross-tokenizer distillation setting, outperforms baselines by utilizing a well-aligned distillation objective. The distilled models maintain strong performance despite reduced vocabulary sizes. On GSM8K, the 32k variant improves over the original by nearly 5% and surpasses SFT by approximately 9%, while reducing the memory footprint by about 12%. These results highlight the advantages of our algorithm and perfect sequence alignment for this problem.

## 7    CONCLUSION

This paper shows cross-tokenizer scoring algorithms that compute sequence likelihoods for any BPE encoding, regardless of the language model's native tokenizer, with finite time termination. For scenarios where the target vocabulary is a subset of the model's vocabulary, our approach delivers exact likelihoods and supports sequential sampling with only $\mathcal{O}(1)$ model evaluations per token. For arbitrary vocabularies, we propose a lossless method that leverages BPE's recursive structure, alongside a fast approximation designed for large vocabularies. Experimental results confirm robust cross-model knowledge distillation across incompatible tokenizers, underscoring the versatility and precision of our algorithms. Future work will focus on optimizing the general-case algorithm and its approximation by reducing time and memory demands through techniques such as enhanced pruning or beam search. Moreover, our approach's flexibility, which avoids intermediate steps like sentence alignment used in other methods, makes it particularly well-suited for modern distillation techniques, such as on-policy distillation (Agarwal et al., 2024). Related problems also include cross-tokenizer preference optimization where the verifier can only output reward for a specific token. Another promising direction involves applying our method to tokenization adaptation (Zheng et al., 2024), where we anticipate significant performance improvements in downstream tasks and target domains, further amplifying the practical impact of our algorithms.

## ACKNOWLEDGMENT

Resources used in preparing this research were provided, in part, by the Province of Ontario, the Government of Canada through CIFAR, and companies sponsoring the Vector Institute www.vectorinstitute.ai/partnerships/.

## REPRODUCIBILITY STATEMENT

To ensure transparency and reproducibility, we will release all artifacts required to replicate our results, including environment specifications, training/evaluation scripts, and code.

## STATEMENT ON THE USE OF LARGE LANGUAGE MODELS

We did not use large language models for any part of this paper (including experiments, analyses, proofs, or writing). All conceptual contributions, study design, implementation, and writing were carried out by the authors.

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

---

**Algorithm 2:** Relative Cover Encoding Search

---

**Input:** Encoding $\vec{e}_{M'}$ over vocabulary $\mathcal{V}_{M'}$; target vocabulary $\mathcal{V}_M$;

1 **Function** RelativeCoverSearch($\vec{e}_{\mathcal{V}_{M'}}, \mathcal{V}_M$):
2     cover_dict = {}
3     $s = \text{decode}(\vec{e}_{\mathcal{V}_{M'}}; M' \to 0)$
4     **for** $i \leftarrow 1$ **to** $|s|$ **do**
5        #Find all tokens with prefix $s[i:]$ and tokenize $s[:i-1]$
6        $\mathcal{B} = \{t \in \mathcal{V}_M | s[i:] \in \text{prefix}(\text{decode}(t))\}$
7        left_tokens $= \text{encode}(s[:i-1]; 0 \to \mathcal{V}_M)$
8        #Filter only valid tokens
9        **for** right_token $\in \mathcal{B}$ **do**
10           $s' = \text{decode}(\text{left\_tokens} \cdot \text{right\_token})$
11           check1 $= \text{is\_valid}(\text{left\_tokens} \cdot \text{right\_token})$
12           check2 $= \text{is\_equal}\big(\vec{e}_{\mathcal{V}_{M'}}; \text{encode}(s'; 0 \to M')\big)[: |\vec{e}_{\mathcal{V}_{M'}}|]$
13           **if** check1 *and* check2 **then**
14              cover_dict.add(left_tokens $\cdot$ right_token)

15     **return** cover_dict

---

# A    Efficient Next-Token Sampling for Full-to-Subset Conversion

## A.1   Relative Cover Encoding

We begin by recalling the definition of relative cover encodings.

**Relative Cover Encodings** Let $\mathcal{V}_{M'} \preceq \mathcal{V}_M$, and let $\vec{e}_{M'}$ be a valid encoding over the vocabulary $\mathcal{V}_{M'}$. We say that an encoding $\vec{e}_{\mathcal{V}_M}$ over $\mathcal{V}_M$ is a *relative cover encoding* of $\vec{e}_{\mathcal{V}_{M'}}$ if the following conditions hold:

1. There exists a sequence $\vec{x}$ such that $\vec{e}_{\mathcal{V}_M} = \text{encode}(\vec{x}; M)[: |\vec{e}_{\mathcal{V}_M}|]$.

2. There exists an index $i \leq |\vec{e}_{\mathcal{V}_{M'}}|$ such that:
$$\text{decode}_{M \to M'}(\vec{e}_{\mathcal{V}_M}[1:|\vec{e}_{\mathcal{V}_M}| - 1]) = \vec{e}_{\mathcal{V}_{M'}}[1:i-1],$$
$$\vec{e}_{\mathcal{V}_{M'}}[i:] \in \text{prefix}\left(\text{decode}_{M \to M'}(\vec{e}_{\mathcal{V}_M}[-1])\right),$$
     i.e. the last token of $\vec{e}_{\mathcal{V}_M}$ starts from some location within $\vec{e}_{\mathcal{V}_{M'}}$.

In BPE, any $\vec{e}_{\mathcal{V}_M}$ must be a valid encoding according to the first condition. Algorithm 2 leverages this by first converting $\vec{e}_{\mathcal{V}_{M'}}$ to its string form $s$. Because the final token of a valid cover must match a suffix of $s$, we traverse $s$ left-to-right: at each position $i$, select the candidates right_token $\in \mathcal{V}_M$ whose decoded string is a prefix of $s[i:]$, and concatenate it after left_tokens $\leftarrow \text{encode}\big(s[:i]\big)$ to form a candidate cover. We then set $s' \leftarrow \text{decode}\big(\text{left\_tokens} \cdot \text{right\_token}\big)$, re-encode $s'$ under the sub-vocabulary to obtain $\vec{e}'_{\mathcal{V}_{M'}} \leftarrow \text{encode}(s'; 0 \to M')$, and verify that the resulting encoding begins with $\vec{e}_{\mathcal{V}_{M'}}$.

## A.2   Next Sub-Token Sampling

In this section, we focus on computing the next-token probability for the subset vocabulary $\mathcal{V}_{M'}$ using the LM operating on the original vocabulary $\mathcal{V}$, i.e. $P_M(.|\vec{e}_{M'})$, similar to the next-byte sampling procedure in (Phan et al., 2025). We assume that we have access to:

1. The set of (relative) cover encoding and the corresponding probabilities of $\vec{e}_{M'}[: |\vec{e}_{M'}| - 1]$, i.e. exclude the last token, obtained either from Algorithm 2 or from previous sampling call (to be shown).

2. The conditional next-token probability distribution $P_M(.|\text{encode}_{M' \to M}(\vec{e}_{M'}[: |\vec{e}_{M'}| - 1]))$, obtained via previous model inference call.

We define the following indicator function for the event that the token $t \in \mathcal{V}_M$ begins with the sub-token $t' \in \mathcal{V}'_M$ through the $\mathrm{decode}_{M \to M'}(\cdot)$ operator.

$$\tau(t, t') \: : \: \mathcal{V}_M \times \mathcal{V}_{M'} \to \{0, 1\}, \qquad \tau(t, t') \triangleq \begin{cases} 1, & \text{if } \mathrm{decode}_{M \to M'}(t)[0] = t', \\ 0, & \text{otherwise}, \end{cases} \tag{14}$$

We can proceed the sampling process through the follow 2-step procedure:

**Step 1 (Relative Cover-Encodings Construction).** We construct the cover encoding of $\vec{\mathbf{e}}_{M'}$ by:

1. Remove the encodings in $\mathrm{cover}(\vec{\mathbf{e}}_{M'}[: |\vec{\mathbf{e}}_{M'}| - 1])$ whose last token $t$ does not begins with the sampled sub-token $\vec{\mathbf{e}}_{M'}[-1]$, i.e. $\tau(t, \vec{\mathbf{e}}_{M'}[-1]) = 0$. We call this set $C_{\mathrm{new}}$.
2. Add the valid encodings $\mathrm{encode}_{M' \to M}(\vec{\mathbf{e}}_{M'}[: |\vec{\mathbf{e}}_{M'}| - 1]) \cdot t$ for any tokens $t$ in $\mathcal{V}_M$ that *begins* with the sampled sub-token $\vec{\mathbf{e}}_{M'}[-1]$, i.e. $\tau(t, \vec{\mathbf{e}}_{M'}[-1]) = 1$, in $C_{\mathrm{new}}$.

The resulting set $C_{\mathrm{new}}$ thus is the desired $\mathrm{cover}(\vec{\mathbf{e}}_{M'})$.

**Step 2 (Compute Conditional Distribution).** To sample the next token $t' \in \mathcal{V}_{M'}$ according $P_{M'}(. | \vec{\mathbf{e}}_{M'})$, we note that:

$$P_{M'}(\vec{\mathbf{e}}_{M'} \cdot t') = P_{M'}(\vec{\mathbf{e}}_{M'} \cdot t', \delta(\vec{\mathbf{e}}_{M'} \cdot t') = 1) + P_{M'}(\vec{\mathbf{e}}_{M'} \cdot t', \delta(\vec{\mathbf{e}}_{M'} \cdot t') = 0), \tag{15}$$

where $\delta(\vec{\mathbf{e}}_{M'} \cdot t') = 1$ if the sub-vocab encoding $\vec{\mathbf{e}}_{M'} \cdot t' \in \mathrm{decode}_{M \to M'}(\vec{\mathbf{e}'}_M)$ where $\vec{\mathbf{e}'}_M \in \mathrm{cover}(\vec{\mathbf{e}}_{M'})$ and 0 vice versa. In other words, it is an indicator function whether the sub-vocab encoding is contained within one of the cover encoding of $\vec{\mathbf{e}}_{M'}$. Note that the first term is the sum of all cover encoding probabilities that satisfy the indicator function property.

For the second term, we have:

$$P_{M'}(\vec{\mathbf{e}}_{M'} \cdot t', \delta(\vec{\mathbf{e}}_{M'} \cdot t') = 0) \tag{16}$$
$$= P_M(t \in \mathcal{V}_M \text{ starts with } t' | \mathrm{encode}_{M' \to M}(\vec{\mathbf{e}}_{M'})) P_M(\mathrm{encode}_{M' \to M}(\vec{\mathbf{e}}_{M'})), \tag{17}$$

where the first term in the product is the model evaluation step we need to run and the second term is already computed from previous cover encodings step. We note that computing the first term in Equation (17) can be done in parallel. In particular, we can pre-compute the fixed prefix matrix $Q = \mathbb{R}^{|\mathcal{V}_{M'}| \times |\mathcal{V}_M|}$ where:

$$Q[i, j] = \begin{cases} 1, & \text{if } t' = \mathcal{V}_{M'}[i] \text{ is a prefix of } t = \mathcal{V}_M[j], \\ 0, & \text{otherwise}. \end{cases} \tag{18}$$

and thus for the sub-token $t' = \mathcal{V}_{M'}[i]$, we have:

$$P_M(t \in \mathcal{V}_M \text{ starts with } t' | \mathrm{encode}_{M' \to M}(\vec{\mathbf{e}}_{M'})) \tag{19}$$
$$= Q \cdot P_M(\cdot | \mathrm{encode}_{M' \to M}(\vec{\mathbf{e}}_{M'}))[i], \tag{20}$$

where $P_M(\cdot | \mathrm{encode}(\vec{\mathbf{e}}_{M'}; M' \to M))$ can be computed in one inference round. Thus this sampling procedure requires only 1 inference step per next token. We show the whole procedure in Algorithm 3.

## B  PROOF OF LEMMA 1

**Lemma 1.** Let $\mathcal{V}_{M'} \preceq \mathcal{V}_M$, and let $P(\cdot)$ be a language model over any arbitrary vocabulary $\mathcal{V}$ with $\mathcal{A} \subset \mathcal{V}$. Given a prefix encoding $\vec{\mathbf{e}}_{\mathcal{V}_{M'}}$ over $\mathcal{V}_{M'}$, the probability that the language model $P(.)$ generates a string whose encoding under $\mathcal{V}_{M'}$ begins with $\vec{\mathbf{e}}_{\mathcal{V}_{M'}}$ is given by:

$$P(\vec{\mathbf{e}}_{\mathcal{V}_{M'}}) = \sum_{\vec{\mathbf{e}}_{\mathcal{V}_M} \in \mathcal{C}} P(\vec{\mathbf{e}}_{\mathcal{V}_M}), \tag{21}$$

where $\mathcal{C} \triangleq \mathrm{cover}_{M' \to M}(\vec{\mathbf{e}}_{\mathcal{V}_{M'}})$ is the set of relative cover encodings in Definition 4.

*Proof.* This result follows as a corollary of the Byte-Token Representation Lemma in Phan et al. (2025), where the vocabulary $\mathcal{V}_{M'}$ is treated as a relative alphabet with respect to $\mathcal{V}_M$. The summation over the cover set arises naturally from marginalizing over all tokenizations in $\mathcal{V}_M$ that map to the same decoded prefix under $\mathcal{V}_{M'}$, as established in Section 3.2. □

---

**Algorithm 3:** Sampling Next Sub-Token

---

**Input:** Encoding $\vec{e}_{M'}$ over vocabulary $\mathcal{V}_{M'}$; target vocabulary $\mathcal{V}_M$;
   Relative Cover Sets $\{(c_i, p_i)\} = \text{cover}_{M' \to M}(\vec{e}_{\mathcal{V}_{M'}}[: |\vec{e}_{M'}| - 1])$ (including the probabilities $p_i$ for each cover $c_i$);
   Conditional distribution $p_{\text{cond}}(\cdot) = P_M\big(\cdot \mid \text{encode}_{M' \to M}(\vec{e}_{M'}[: |\vec{e}_{M'}| - 1])\big)$.

1  **Function** Next_SubToken_Sampling($\vec{e}_{\mathcal{V}_{M'}}, \mathcal{V}_M, \{(c_i, p_i)\}, p_{\text{cond}}(\cdot)$):
2      **Step 1: Relative Cover-Encodings Construction**;
3      $\text{cover}_{M' \to M}(\vec{e}_{\mathcal{V}_{M'}}) \leftarrow$ Cover_Construction($\vec{e}_{\mathcal{V}_{M'}}, \mathcal{V}_M, \{(c_i, p_i)\}, p_{\text{cond}}(\cdot)$);
4      **Step 2: Compute Conditional Distribution**;
        // Note:    $\text{encode}_{M' \to M}(\vec{e}_{M'}), P_M(\text{encode}_{M' \to M}(\vec{e}_{M'})) \in \text{cover}_{M' \to M}(\vec{e}_{\mathcal{V}_{M'}})$
5      $p_{\text{next}}(\cdot) \leftarrow Q \cdot P_M\big(\cdot \mid \text{encode}_{M' \to M}(\vec{e}_{M'})\big) \cdot P_M\big(\text{encode}_{M' \to M}(\vec{e}_{M'})\big)$;
6      **for** $(c, p) \in \text{cover}_{M' \to M}(\vec{e}_{\mathcal{V}_{M'}})$ **do**
7          $u' \leftarrow \text{decode}_{M \to M'}(c)[|\vec{e}_{M'}| :]$;
8          $t' \leftarrow u'[0]$;
9          $p_{\text{next}}[t'] \leftarrow p_{\text{next}}[t'] + p$;
10     $p_{\text{next}}(\cdot) \leftarrow p_{\text{next}}(\cdot) / \sum p_{\text{next}}(\cdot)$;
11     Sample next-token $t' \sim p_{\text{next}}(\cdot)$;
12     **return** $p(\cdot), \text{cover}_{M' \to M}(\vec{e}_{\mathcal{V}_{M'}})$;

13 **Function** Cover_Construction($\vec{e}_{\mathcal{V}_{M'}}, \mathcal{V}_M, \{(c_i, p_i)\}, p_{\text{cond}}(\cdot)$):
        // Recall:    $\text{cover}_{M' \to M}(\vec{e}_{\mathcal{V}_{M'}}[: |\vec{e}_{M'}| - 1]) = \{(c_i, p_i)\}$
14     $C_{\text{new}} \leftarrow \{\}$;
15     **for** $(c, p) \in \text{cover}_{M' \to M}(\vec{e}_{\mathcal{V}_{M'}}[: |\vec{e}_{M'}| - 1])$ **do**
16         $t \leftarrow c[-1]$;
17         **if** $\tau(t, \vec{e}_{M'}[-1]) = 1$ **then**
18             $C_{\text{new}}.\text{append}((c, p))$;
19     $\vec{b}_M \leftarrow \text{encode}_{M' \to M}\big(\vec{e}_{M'}[: |\vec{e}_{M'}| - 1]\big)$;
20     **for** $t \in \mathcal{V}_M$ such that $\tau(t, \vec{e}_{M'}[-1]) = 1$ **and** is_valid($\vec{b}_M \cdot t$) **do**
21         $\vec{y}_M \leftarrow \vec{b}_M \cdot t$;
22         $P_M(\vec{y}_M) = \big(p_{\text{cond}} \cdot P_M(\vec{b}_M)\big)[t]$;
23         $C_{\text{new}}.\text{append}((\vec{y}_M, P_M(\vec{y}_M)))$;
24     **return** $C_{\text{new}}$;

---

## C    Proof of Algorithm 1

To prove the recursion step, we characterize the transition between adjacent vocabularies.

**Lemma 2** (Adjacent Vocabulary Transition). *Let $\mathcal{V}_{M'} \preceq \mathcal{V}_{M'+1}$ be adjacent vocabularies, and let $P_{M'}(.)$ denote a language model defined over $\mathcal{V}_{M'}$. Let $\vec{e}_{\mathcal{V}_{M'+1}}$ be an encoding over $\mathcal{V}_{M'+1}$, and define its decoded form under $\mathcal{V}_{M'}$ as*

$$\vec{e}_{\mathcal{V}_{M'}} = \text{decode}_{M'+1 \to M'}(\vec{e}_{\mathcal{V}_{M'+1}}).$$

*Let $t_{M'+1} = t_{M'+1}^{\text{left}} \cdot t_{M'+1}^{\text{right}}$ be the token added in $\mathcal{V}_{M'+1}$. Then, the probability of $\vec{e}_{\mathcal{V}_{M'+1}}$ under $P_{M'}$ is given by:*

$$P_{M'}(\vec{e}_{\mathcal{V}_{M'+1}}) = \begin{cases} P_{M'}\left(\vec{e}_{\mathcal{V}_{M'}}\right) & \text{if } \vec{e}_{\mathcal{V}_{M'+1}}[-1] \neq t_{M'+1}^{\text{left}}, \\ P_{M'}\left(\vec{e}_{\mathcal{V}_{M'}}\right) - P_{M'}\left(\vec{e}_{\mathcal{V}_{M'}} \cdot t_{M'+1}^{\text{right}}\right) & \text{otherwise.} \end{cases}$$

*Proof.* Consider first the case where $\vec{e}_{\mathcal{V}_{M'+1}}[-1] \neq t_{M'+1}^{\text{left}}$. In this case, the relative cover set is

$$\text{cover}_{M' \to M'+1}(\vec{e}_{\mathcal{V}_{M'}}) = \{\vec{e}_{\mathcal{V}_{M'+1}}\},$$

since the last token $\vec{\mathbf{e}}_{\mathcal{V}_{M'+1}}[-1]$ is a part of any merged token in $\mathcal{V}_{M'+1}$, including $t_{M'+1}$. Hence, the probability under $P_{M'}$ remains unchanged:

$$P_{M'}(\vec{\mathbf{e}}_{\mathcal{V}_{M'+1}}) = P_{M'}(\vec{\mathbf{e}}_{\mathcal{V}_{M'}}).$$

Now consider the case where $\vec{\mathbf{e}}_{\mathcal{V}_{M'+1}}[-1] = t_{M'+1}^{\text{left}}$. In this case, the relative cover set becomes

$$\text{cover}_{M' \to M'+1}(\vec{\mathbf{e}}_{\mathcal{V}_{M'}}) = \left\{ \vec{\mathbf{e}}_{\mathcal{V}_{M'+1}}, \ \vec{\mathbf{e}}_{\mathcal{V}_{M'+1}}[1:|\vec{\mathbf{e}}_{\mathcal{V}_{M'+1}}|-1] \cdot t_{M'+1} \right\}.$$

Applying Lemma 1, which decomposes the marginal probability over all possible relative cover encodings, we obtain:

$$P_{M'}(\vec{\mathbf{e}}_{\mathcal{V}_{M'+1}}) = P_{M'}(\vec{\mathbf{e}}_{\mathcal{V}_{M'}}) - P_{M'}\left( \vec{\mathbf{e}}_{\mathcal{V}_{M'+1}}[1:|\vec{\mathbf{e}}_{\mathcal{V}_{M'+1}}|-1] \cdot t_{M'+1} \right) \tag{22}$$

$$= P_{M'}(\vec{\mathbf{e}}_{\mathcal{V}_{M'}}) - P_{M'}\left( \vec{\mathbf{e}}_{\mathcal{V}_{M'}} \cdot t_{M'+1}^{\text{right}} \right), \tag{23}$$

where the second equality is due to the previous case, i.e. $t_{M'+1} \neq t_{M'+1}^{\text{left}}$ is not a part of any other token in $\mathcal{V}_{M'+1}$. This completes the proof. $\qquad\square$

Finally, given the result in Lemma 2 the recursion step in Algorithm 1 follows naturally.

### C.1 SCORING AND SAMPLING WITH BEAM-SEARCH

**Approximate Scoring Procedure.** We focus on the conversion case from the byte-level model $\mathcal{V}_0 = \mathcal{A}$ to a target vocabulary $\mathcal{V}_M$, denoted simply as $\mathcal{V}$. Let $\vec{\mathbf{e}}$ be an encoding in $\mathcal{V}$ with string representation $s = \text{decode}(\vec{\mathbf{e}})$. From the lossless Algorithm 1, we have

$$P_{\mathcal{V}}(\vec{\mathbf{e}}) = P_0(s) - \sum_{s' \in \Omega} P_0(s'), \tag{24}$$

where $P_0(\cdot)$ denotes probabilities under the byte-level model, and $\Omega$ is the correction set returned by Algorithm 1. Each $P_0(s')$ represents the probability that a sequence begins with prefix $s'$. Since evaluating the full set $\Omega$ is generally intractable, our goal is to approximate the correction term $\sum_{s' \in \Omega} P_0(s')$ by identifying a restricted collection of candidate strings $s'$ that capture the dominant contributions, without explicitly enumerating all elements of $\Omega$. First, observe that

$$P_0(s) = P\big(\text{sequence starts with } s \text{ and matches } \vec{\mathbf{e}}\big) + P\big(\text{sequence starts with } s \text{ but not } \vec{\mathbf{e}}\big) \tag{25}$$

$$= P_{\mathcal{V}}(\vec{\mathbf{e}}) + P\big(\text{sequence starts with } s \text{ but not } \vec{\mathbf{e}}\big), \tag{26}$$

and hence

$$P_{\mathcal{V}}(\vec{\mathbf{e}}) = P_0(s) - P\big(\text{sequence starts with } s \text{ but not } \vec{\mathbf{e}}\big). \tag{27}$$

We now make the following approximation:

$$P\big(\text{sequence starts with } s \text{ but not } \vec{\mathbf{e}}\big) \approx \sum_{s' \in \Pi} P(s'), \tag{28}$$

where:

$$\Pi = \{ s' \text{ starts with } s, \tag{29}$$

$$s'[-1] \text{ is the first white space/EOS appears after prefix } s, \tag{30}$$

$$\text{encode}(s')[:|\vec{\mathbf{e}}|] \neq \vec{\mathbf{e}}\} \tag{31}$$

This assumption is not overly restrictive, particularly for languages such as English where whitespace is frequently used. Intuitively, it asserts that a continuation $s'$ of $s$ is more likely to correspond to a valid "word" rather than an arbitrary long character sequence. Moreover, under the standard pre-tokenization rule in which whitespace appears only at the beginning of a token, once a whitespace is sampled, we can uniquely determine which encoding of $s'$ follows up to that point [3].

---

[3] Readers may find this resembles—though is not identical to—the process of computing the probability of a word (Pimentel & Meister, 2024).

Combine all the terms, we have the following approximation for $P_\mathcal{V}(\vec{\mathbf{e}})$:

$$P_\mathcal{V}(\vec{\mathbf{e}}) \approx P_0(s) - \sum_{s' \in \Pi} P(s'), \tag{32}$$

We summarize the procedure in two steps. Assume access to a byte-level LM $P_0(\cdot)$ (given or obtained via conversion), and let $\vec{\mathbf{e}}$ denote the target encoding.

1. *Beam-search:* Starting from $P_0(.|s)$, generates $N$ candidate beams $s'_1, ...s'_N$ using beamsearch until seeing a whitespace/EOS.
2. *Return:* $P_\mathcal{V}(\vec{\mathbf{e}}) \approx P_0(s) - \sum_{s' \in \Pi'} P(s')$.

**Sampling a new token.** Given a byte-level language model $P_0$ and current encoding $\vec{\mathbf{e}}$, we perform byte-level sampling autoregressively until a stopping criterion is met (whitespace or a maximum length of a cover encoding, ...). Let $s$ be the sampled string and $\text{encode}(s)$ its tokenization under $\mathcal{V}$. If $\text{encode}(s)$ begins with $\vec{\mathbf{e}} \cdot t$ for some $t \in \mathcal{V}$, output $t$; otherwise, reject $s$ and resample without recommitting to that path.

# D    DISTILLATION LOSS

We employ the following two distillation objectives for transferring knowledge from a teacher language model $P_\text{teacher}$ to a student model $P_\text{student}$. In our experiment, we find that combining distillation loss with the SFT loss gives better results.

**Per-token KL Distillation (Hinton et al., 2015).** Following the standard distillation framework, the teacher provides a full predictive distribution $P_\text{teacher}(\cdot \mid x_{<t})$ over the student vocabulary $\mathcal{V}_\text{student} \triangleq \mathcal{V}$. The student is trained to minimize the token-wise Kullback–Leibler (KL) divergence between the teacher and student distributions:

$$\mathcal{L}_\text{KL} = \sum_{\ell=1}^{L} D_\text{KL}\Big( P_\text{teacher}(\cdot \mid x_{<\ell}) \,\big\|\, P_\text{student}(\cdot \mid x_{<\ell}) \Big) \tag{33}$$

$$= -\sum_{\ell=1}^{L} \sum_{t \in \mathcal{V}} P_\text{teacher}(t \mid x_{<\ell}) \log \frac{P_\text{student}(t \mid x_{<\ell})}{P_\text{teacher}(t \mid x_{<\ell})}. \tag{34}$$

This loss encourages the student to match the teacher's complete predictive distribution at every position. It is particularly suitable for the sub-vocabulary distillation problem since we can obtain the probability distribution over the student vocabulary in $\mathcal{O}(1)$ model inference.

**Partial KL Divergence Loss.** In general, cross-tokenizer distillation, obtaining the full probability distribution over the vocabulary is impractical due to the time-intensive process of calculating each token's likelihood (approximately 0.5 seconds per token). In our approach, we typically compute probabilities for about three cross-(next) tokens per encoded prompt, leaving a significant portion of the probability mass unexamined. For this reason, we adjust the above KL divergence loss to take into account the unknown probability mass. In particular, we optimize:

$$\mathcal{L}_\text{bin} = -\sum_{\ell=1}^{L} \Big[ \sum_{t \in Q(\ell)} q_t \log P_S(y_t \mid x_{<t}) + \big(1 - \sum_{t \in Q(\ell)} q_t\big) \log \big(1 - \sum_{t \in Q(\ell)} P_S(y_t \mid x_{<t})\big) \Big], \tag{35}$$

where $Q(\ell)$ is the set of random tokens in $\mathcal{V}$ sampled at step $\ell$ by the teacher. In the case $Q(\ell)$ contains only the next token in the corpus, this recovers and generalizes the binary cross entropy loss in (Minixhofer et al., 2025), without employing the sequence alignment step.

| Vocab | Task | SFT | FKL (Ours) | FKL* (Ours) | ALM | ALM* | ULD | ULD* | DSKD | DSKD* |
|---|---|---|---|---|---|---|---|---|---|---|
| 64k | ARC-C | 46.5 | 46.5 | **46.9** | 46.4 | 46.3 | 45.1 | 46.2 | 45.9 | 46.3 |
| | ARC-E | 72.9 | 73.2 | **74.0** | 72.4 | 72.5 | 72.2 | 72.4 | 73.1 | 73.6 |
| | BOOLQ | **79.8** | 77.5 | 78.9 | 78.2 | 79.3 | 75.5 | 75.8 | 77.2 | 78.9 |
| | COMMONSENSE | 75.1 | 75.2 | **75.4** | 75.3 | 74.9 | 75.3 | 75.1 | 74.7 | 75.2 |
| | Hellaswag | 49.2 | 48.9 | **49.5** | 48.9 | 49.1 | 48.1 | 49.0 | 48.5 | 48.8 |
| | PIQA | 75.2 | 74.9 | **75.5** | 74.5 | 75.0 | 75.1 | 75.3 | 72.1 | 74.9 |
| 32k | ARC-C | 43.7 | 43.2 | **44.4** | 43.3 | 43.9 | 42.4 | 44.2 | 43.3 | 44.0 |
| | ARC-E | 71.2 | 68.9 | **71.6** | 67.7 | 68.6 | 66.5 | 67.1 | 69.9 | 70.2 |
| | BOOLQ | 78.3 | **78.8** | 78.5 | 78.3 | 78.6 | 78.4 | 78.2 | 77.2 | 78.6 |
| | COMMONSENSE | 74.5 | 75.0 | **75.7** | 75.1 | 75.0 | 72.4 | 73.4 | 72.9 | 73.8 |
| | Hellaswag | 47.5 | 47.2 | **47.8** | 46.9 | 47.4 | 46.3 | 46.9 | 47.1 | 47.5 |
| | PIQA | 73.2 | 72.3 | **73.5** | 71.9 | 72.2 | 70.1 | 71.0 | 72.5 | 72.9 |
| 16k | ARC-C | 40.1 | **42.4** | 42.0 | 41.4 | 41.8 | 40.1 | 41.8 | 40.9 | 41.2 |
| | ARC-E | 66.9 | 66.8 | **68.1** | 65.1 | 66.6 | 62.9 | 64.1 | 63.5 | 64.7 |
| | BOOLQ | 78.8 | 78.1 | **78.9** | 78.6 | 78.6 | 72.1 | 72.4 | 72.8 | 73.6 |
| | COMMONSENSE | **74.7** | 74.3 | 73.8 | 74.4 | 74.3 | 70.1 | 70.3 | 71.0 | 72.8 |
| | Hellaswag | 45.4 | 45.3 | **45.7** | 44.9 | 45.2 | 40.1 | 42.8 | 41.1 | 43.8 |
| | PIQA | **71.4** | 70.6 | 71.0 | 70.1 | 70.6 | 67.1 | 68.9 | 70.5 | 70.7 |

Table 5: Performance across vocab sizes on common reasoning datasets. We use 4-shot prompting for all tasks and setups. * denotes distillation loss function combined with SFT objective.

# E  ADDITIONAL EXPERIMENT

**Evaluation Frameworks.** All experiments, including the cross-tokenizer distillation, are evaluated using the lm-eval framework (Gao et al., 2024).

**Cross-Tokenizer Distillation.** We additionally evaluate the general cross-tokenizer distillation setting on the DialogSum summarization dataset (Chen et al., 2021), following the experimental setup of (Xu et al., 2025). DialogSum contains 12.5k examples; we use 10k for supervised finetuning of the teacher model (`Qwen2.5-7B-Instruct`), 1k for distilling the student model (`Gemma-2B-Instruct`) using the SFT finetuned teacher, with the remaining 1.5k for validation. This configuration reflects a common low-resource distillation scenario, where only a small student-side dataset is available during knowledge transfer.

| Method | ROUGE-L |
|---|---|
| Gemma-2B-IT (Student) | 12.3 |
| Qwen2.5-7B-IT (Teacher) | 39.1 |
| SFT | 31.9 |
| ULD (Boizard et al., 2025) | 28.1 |
| ULD + SFT (Boizard et al., 2025) | 32.2 |
| DSKD (Zhang et al., 2024) | 30.3 |
| DSKD + SFT (Zhang et al., 2024) | 32.8 |
| ALM | 20.5 |
| ALM + SFT | 33.1 |
| PKL (Ours) | 32.6 |
| SFT + PKL (Ours) | **33.9** |

Table 4: Cross-Tokenizer distillation (Qwen2.5–7B-IT → Gemma-2B-IT) on the the DialogSum dataset.

During the distillation phase, we train the student with a batch size of 8 for a total of 325 steps (3 epochs) using a learning rate of $1 \times 10^{-5}$. Overall, we find that our approach remains competitive with existing techniques and achieves the highest summarization score, highlighting the generality and robustness of our framework across tokenizer mismatches.

**Vocabulary Trimming.** We report additional results on common reasoning benchmarks. In this setting, rather than fine-tuning on each target dataset (as in the GSM8K and coding experiments), we continue training on the Alpaca dataset for two more epochs and then evaluate on the benchmarks. The results is shown in Table 5 where our method consistently outperforms the baseline methods on most benchmarks.

