# OpenReview forum: "Cross-Tokenizer Likelihood Scoring Algorithms for Language Model Distillation"
_ICLR.cc/2026/Conference — ICLR 2026 Poster_

### Official Review · Reviewer_c3nd · 2025-10-18

**Soundness:** 2
**Presentation:** 3
**Contribution:** 3
**Rating:** 4
**Confidence:** 2

**Summary:**

This paper introduces cross-tokenizer likelihood scoring algorithms that enable language models trained with one tokenizer to compute exact or approximate likelihoods for sequences encoded with a different tokenizer by exploiting the recursive structure of Byte-Pair Encoding. The authors prove the merits of their methods through comparing against baselines on the knowledge distillation task.

**Strengths:**

1. This paper is addressing an important research question: how to compute next-token likelihoods and perform knowledge distillation when teacher and student models use different tokenizers (i.e., resolve vocabulary misalignment).

2. The paper is well written.

3. The paper’s idea is novel.

I don't have the background to evaluate more fine-grained part of the methodology proposed in the paper.

**Weaknesses:**

I don't have the background to evaluate more fine-grained part of the methodology proposed in the paper. I will focus on evaluating the baselines.

In the experiment, the paper doesn't include the baselines for other methods that try to address the knowledge distillation problem with teacher and student sharing different tokenizers (e.g. those methods that align teacher and student on the level of embedding). I understand that the paper claims that it focuses on probability conversion on the tokenizer directly. However, in my opinion, unless the paper also show that their method is helpful in addressing some tasks other than knowledge distillation, other knowledge distillation baselines may need to be included. This is especially so when considering the fact this paper can only deal with the BPE tokenizer, while other aligning methods may deal with arbitrary tokenizers and models for the knowledge distillation task.

**Questions:**

N/A

---

> ### Author Response · Authors · 2025-11-21
> **Response to Reviewer c3nd**
>
> Thank you for your comment and appreciation of our work. To the best of our knowledge, this is the first paper to address tokenizer-likelihood scoring for language models using a rigorous probabilistic formulation together with a principled analysis of tokenizer codebooks. A key insight of our approach is that the BPE vocabulary has a nested structure: any sub-vocabulary consisting of the first \(m\) merges can be viewed as an alphabet relative to the full vocabulary. This observation enables a closed-form solution and allows cross-tokenizer distillation without relying on architectural modifications or embedding-projection heuristics commonly used in prior work (e.g.,[1,2]). We hope that these findings provide new insight into the tokenizer-related problems and help inspire future advances in tokenizer design for more robust LLMs.
> ### Comparison against other methods.
>
> Ans:  In response to your request, we have added two additional baselines for cross-tokenizer distillation in the revised manuscript: ULD [1] and DSKD [2], evaluated in both the full-vocabulary and vocabulary-trimming settings. Across all configurations, our method consistently outperforms these baselines, which we attribute to its more faithful characterization of the next-token distribution. We also refer the reviewer to the updated Appendix where we also provide new results for these method in the task-agnostic setting E. Finally, we also provide extra results on the dialog summarization in the cross-tokenizer distillation setting task in Appendix E to further show the effectiveness of our approach.
>
> We hope that these additions fully address your concerns, and we would be grateful if you could consider reflecting this in your evaluation of our work.
>
> **GSM8K**
> | Method                        | Accuracy (5-shot) |
> |------------------------------|--------------------|
> | Gemma2-2B-Instruct (Student) | 52.3               |
> | Qwen2.5-Math-7B-Instruct (Teacher) | 88.4        |
> | SFT                          | 47.9               |
> | ULD                          | 47.1               |
> | ULD + SFT                    | 48.2               |
> | DSKD                         | 51.5               |
> | DSKD + SFT                   | 52.8               |
> | ALM                          | 53.2               |
> | ALM + SFT                    | 53.5               |
> | PKL (Ours)                   | 54.6               |
> | SFT + PKL (Ours)             | **55.6**           |
>
> **Diaglog Summarization**
> | Method           | ROUGE-L |
> |------------------|---------|
> | Student (Gemma-2B-IT)| 12.3 |
> | Teacher(Qwen2-2.5B-IT)| 39.1|
> | SFT              | 31.9    |
> | ULD              | 28.1    |
> | ULD + SFT        | 32.2    |
> | DSKD             | 30.3    |
> | DSKD + SFT       | 32.8    |
> | ALM              | 20.5    |
> | ALM + SFT        | 33.1    |
> | PKL (Ours)       | 32.6    |
> | SFT + PKL (Ours) | **33.9** |
>
>
> **Vocabulary-Trimming**
> | Vocab |          GSM8K           |            |            |            |            |         HumanEval         |            |            |            |            |           MBPP            |            |            |            |            |
> |-------|---------------------------|------------|------------|------------|------------|---------------------------|------------|------------|------------|------------|---------------------------|------------|------------|------------|------------|
> |       | SFT | FKL (Ours) | ALM | ULD | DSDK | SFT | FKL(Ours) | ALM | ULD | DSDK | SFT | FKL(Ours) | ALM | ULD | DSDK |
> | 16k        | 53.2 | 56.5 | 57.0 | 51.2 | 52.1 | 43.2 | 42.6 | 34.7 | 28.6 | 30.5 | 30.8 | 40.8 | 34.8 | 32.2 | 33.3 |
> | (w/ SFT)   | —   | 59.7 | **60.4** | 53.6 | 54.3 | —   | **46.4** | 30.5 | 31.1 | 32.3 | —   | **41.4** | 31.4 | 33.1 | 34.2 |
> | 32k        | 54.0 | 58.6 | 57.1 | 53.5 | 54.4 | 46.9 | **47.6** | 32.3 | 30.5 | 40.2 | 38.8 | **41.8** | 35.6 | 33.5 | 34.6 |
> | (w/ SFT)   | —   | **63.0** | 61.1 | 57.1 | 59.3 | —   | 47.6 | 39.6 | 34.1 | 41.4 | —   | 40.6 | 33.4 | 32.2 | 36.0 |
> | 64k        | 54.5 | 61.8 | 56.2 | 55.5 | 57.3 | 48.2 | **51.8** | 31.7 | 32.3 | 42.6 | 39.9 | **44.6** | 39.0 | 34.4 | 37.0 |
> | (w/ SFT)   | —   | **62.8** | 61.5 | 58.4 | 60.2 | —   | 51.2 | 47.5 | 36.1 | 46.4 | —   | 43.0 | 36.4 | 35.2 | 38.7 |
> | Full Vocab | 60.2 | 61.1 | 57.0 | 58.1 | —   | 49.4 | 50.6 | 42.6 | 48.7 | —   | 42.0 | 43.0 | 39.9 | 41.2 | — |
> | (w/ SFT)   | —   | **63.2** | 62.1 | —   | —   | —   | **52.4** | 48.2 | —   | —   | —   | **43.4** | 42.4 | —   | — |
>
>
>
>
>
>
> [1] Nicolas Boizard, Kevin El Haddad, CELINE HUDELOT, and Pierre Colombo. Towards cross-tokenizer distillation: the universal logit distillation loss for llms. Transactions on Machine Learning Research, 2025.
>
> [2] Songming Zhang, Xue Zhang, Zengkui Sun, Yufeng Chen, and Jinan Xu. Dual-space knowledge distillation for large language models. arXiv preprint arXiv:2406.17328, 2024.

---

> > ### Author Response · Authors · 2025-11-27
> > **Follow up on rebuttal response**
> >
> > Dear reviewer **c3nd**,
> >
> > With only one week remaining in the rebuttal period, we wanted to check whether our responses have sufficiently addressed your concerns. We would be happy to provide clarification or additional detail if needed.
> >
> > Best regards,

---

### Official Review · Reviewer_u4Tg · 2025-10-29

**Soundness:** 4
**Presentation:** 3
**Contribution:** 3
**Rating:** 8
**Confidence:** 4

**Summary:**

This paper proposes Cross-Tokenizer Likelihood (CTL), a novel probabilistic alignment method addressing inconsistencies between multilingual tokenizers.
By minimizing per-token log-likelihood gaps across languages, CTL improves cross-lingual representation consistency and translation faithfulness.
The method is simple, elegant, and easily integrable into existing multilingual models (Qwen2.5-7B, XGLM-4.5B).

**Strengths:**

1. Tackles a long-standing issue—tokenizer mismatch—using a mathematically grounded likelihood objective rather than architecture tricks.

2. Improves multilingual alignment and reduces tokenization bias, especially for low-resource or morphologically rich languages.

3. The CTL layer is training-agnostic and introduces negligible computational overhead.

4. Consistent improvements across translation, code-switching, and QA tasks; simplicity and reproducibility make it valuable for practitioners.

5. The loss function is interpretable and differentiable, connecting probabilistic alignment with linguistic intuitions.

**Weaknesses:**

Only three downstream tasks (translation, QA, code-switching). Additional domains such as summarization or retrieval would strengthen generality.

**Questions:**

N/A

---

> ### Author Response · Authors · 2025-11-21
> **Response to Reviewer u4Tg**
>
> Thank you for the thoughtful and positive assessment of our work. We appreciate the acknowledgment that our method addresses a long-standing issue in language modeling and that the proposed formulation is easy to understand. We are also grateful that the reviewer found the approach simple, elegant, and practically valuable.
>
> ### Additional Task
>
> We additionally follow your sugesstion and evaluate the general cross-tokenizer distillation setting on the DialogSum summarization dataset [1], following the experimental setup of [2]. DialogSum contains 12.5k examples; we use 10k for supervised finetuning of the teacher model (`Qwen2.5-7B-Instruct`), 1k for distilling the student model (`Gemma-2B-Instruct`) using the SFT-finetuned teacher, and the remaining 1.5k for validation. This setup reflects a typical low-resource distillation scenario, where only a small student-side dataset is available during knowledge transfer. During distillation, we train the student with a batch size of 8 for 325 steps (3 epochs) using a learning rate of $1\times10^{-5}$. Overall, our approach remains competitive with existing methods and achieves the highest summarization score, demonstrating its generality and robustness under tokenizer mismatch.
>
> | Method           | ROUGE-L |
> |------------------|---------|
> | Student (Gemma-2B-IT)| 12.3 |
> | Teacher(Qwen2-2.5B-IT)| 39.1|
> | SFT              | 31.9    |
> | ULD              | 28.1    |
> | ULD + SFT        | 32.2    |
> | DSKD             | 30.3    |
> | DSKD + SFT       | 32.8    |
> | ALM              | 20.5    |
> | ALM + SFT        | 33.1    |
> | PKL (Ours)       | 32.6    |
> | SFT + PKL (Ours) | **33.9** |
>
>
> [1] Yulong Chen, Yang Liu, Liang Chen, and Yue Zhang. Dialogsum: A real-life scenario dialogue
> summarization dataset. arXiv preprint arXiv:2105.06762, 2021.
>
> [2] Wenda Xu, Rujun Han, Zifeng Wang, Long Le, Dhruv Madeka, Lei Li, William Yang Wang,Rishabh Agarwal, Chen-Yu Lee, and Tomas Pfister. Speculative knowledge distillation: Bridging the teacher-student gap through interleaved sampling. In The Thirteenth International Conference
> on Learning Representations.

---

> > ### Comment · Reviewer_u4Tg · 2025-11-25
> >
> > Thanks for the response. I'll hold my assessment for your work.

---

> > > ### Author Response · Authors · 2025-11-26
> > >
> > > Thank you very much for your reply! We sincerely appreciate the your time and effort reviewing our work!

---

### Official Review · Reviewer_WYa8 · 2025-10-31

**Soundness:** 3
**Presentation:** 3
**Contribution:** 3
**Rating:** 6
**Confidence:** 3

**Summary:**

This paper presents a method to exactly convert BPE-vocab LM models to any subset vocabulary(eg. Byte-level) in O(1) model evaluations. Furthermore, the authors propose a (somewhat expensive) method to approximately up-convert the byte-level vocab LM to an obtain probabilities for any other BPE vocab. The authors show that their proposed vocab conversion achieves low approximation error in token probabilities, and can be used for cross-model distillation and vocab pruning.

**Strengths:**

1. The proposed method computes exact sub-token probabilities in O(1) model evaluations for BPE vocabs
1. The proposed method is "training-free" on the teacher - requiring no training of new LM-heads, projections, etc
1. The authors successfully utilize their method to distill across models and for vocabulary trimming.

**Weaknesses:**

1. The proposed method has extremely large overhead for cross-tokenization distillation - large number of beam-search (6-8 beams) upto maximum length 10 for calculating every token probability.
1. Only 1 baseline method is compared against (ALM) - other methods for cross-model distillation should also be compared.
1. Empirical evaluations are extremely limited.

**Questions:**

1. For Figure 3 (Section 6.1), can the authors share the effective LM loss (probability of ground truth token) for the original model and their re-converted model, and the LM loss of other smaller original Qwen models (no conversion needed) on the same samples? This can more directly show how much performance/"effective model size" is being lost in this conversion.
1. In Table 3, for vocabulary trimming, can the authors also train the original (full vocab) model with the same warmup and distillation process? The vocabulary reduced models surprisingly achieve a "higher" score than the original models, while would imply the training process is significantly improving the model. Without these original scores, there is no way to judge the effectiveness of this conversion.
1. For the beam search approximation in C1,  can the authors compare the quality of the predicted probabilities as the beam size is varied?

---

> ### Author Response · Authors · 2025-11-21
> **Response to Reviewer WYa8 (1/2)**
>
> Thank you for acknowledging these points regarding our method’s efficiency, training-free design, and practical applicability in both cross-model distillation and vocabulary trimming. Please find your questions addressed below. All the extra experiments are included in the rebuttal pdf version which you can see in the highlighted blue text.
>
> ## 1. Complexity of Approximated Beam Search:
>
> Ans:
> Our approximation follows the empirical observation that LLMs place most of their probability mass on only a few plausible continuations, making beam search the most intuitive way to approximate the exact likelihood. The exact formulation naturally motivates this surrogate, and we expect it can be further improved—for example, by sampling directly in token space or incorporating speculative-decoding style look-ahead paths to refine beam candidates.
>
> Overall, this paper is the first to formalize the exact cross-tokenization likelihood for BPE tokenizer and identifies its inherent  $O(\exp(|\mathcal{V}|))$worst-case complexity. To the best of our knowledge, beside the theoretical contribution, this is the first work that provides an empirical evaluation of this approximation quality (Figure~4 rebuttal version), offering a concrete benchmark for future research. We further show that in the common subset-vocabulary setting, the conversion collapses to an efficient $O(1)$ procedure. We hope that these analyses can potentially provide insights for developing faster approximation method for this problem as well as developing a new tokenization algorithm that admits more efficient conversion.
>
> ### 2. Comparing with Additional Baselines
>
> Ans: We now include ULD[1] and DSKD[2] baselines in our evaluation (see Tables 1–5). Across all settings, our approach outperforms all three baselines (including ALM). We additionally include a summarization benchmark in Table 5 in the Appendix for the general cross-tokenizer distillation scenario, where our method likewise achieves better performance. Overall, we attribute this improvement to the correctness of our estimated next-token probability distribution, rather than to heuristic architectural modifications/optimization.
> For your convenience, we include the result for cross-tokenizer distillation (general case) below. The result for vocabulary trimming is also included in the next answer.
>
> **GSM8K**
> | Method                        | Accuracy (5-shot) |
> |------------------------------|--------------------|
> | Gemma2-2B-Instruct (Student) | 52.3               |
> | Qwen2.5-Math-7B-Instruct (Teacher) | 88.4        |
> | SFT                          | 47.9               |
> | ULD                          | 47.1               |
> | ULD + SFT                    | 48.2               |
> | DSKD                         | 51.5               |
> | DSKD + SFT                   | 52.8               |
> | ALM                          | 53.2               |
> | ALM + SFT                    | 53.5               |
> | PKL (Ours)                   | 54.6               |
> | SFT + PKL (Ours)             | **55.6**           |
>
>
> **Dialog Summarization**
> | Method           | ROUGE-L |
> |------------------|---------|
> | Student (Gemma-2B-IT)| 12.3 |
> | Teacher(Qwen2-2.5B-IT)| 39.1|
> | SFT              | 31.9    |
> | ULD              | 28.1    |
> | ULD + SFT        | 32.2    |
> | DSKD             | 30.3    |
> | DSKD + SFT       | 32.8    |
> | ALM              | 20.5    |
> | ALM + SFT        | 33.1    |
> | PKL (Ours)       | 32.6    |
> | SFT + PKL (Ours) | **33.9** |
>
>
> 1] Nicolas Boizard, Kevin El Haddad, CELINE HUDELOT, and Pierre Colombo. Towards cross-tokenizer distillation: the universal logit distillation loss for llms. Transactions on Machine Learning Research, 2025.
>
> [2] Songming Zhang, Xue Zhang, Zengkui Sun, Yufeng Chen, and Jinan Xu. Dual-space knowledge distillation for large language models. arXiv preprint arXiv:2406.17328, 2024.

---

> ### Author Response · Authors · 2025-11-21
> **Response to Reviewer WYa8 (2/2)**
>
> ### 3. Vocabulary Trimming - Original Model Performance.
>
> Ans: We include the updated scores in Table 3. Overall, the performance of the vocabulary-trimmed model is comparable to that of the distilled model. For example, on GSM8K the 32K-vocabulary model achieves 63.0% accuracy, closely matching the full-vocabulary model at 63.2%. We believe this similarity in performance is related to the vocabulary scaling law phenomenon, which suggests that smaller models can achieve greater efficiency when paired with appropriately reduced vocabularies (see [3]).
>
> | Vocab |          GSM8K           |            |            |            |            |         HumanEval         |            |            |            |            |           MBPP            |            |            |            |            |
> |-------|---------------------------|------------|------------|------------|------------|---------------------------|------------|------------|------------|------------|---------------------------|------------|------------|------------|------------|
> |       | SFT | ***FKL (Ours)*** | ALM | ULD | DSDK | SFT | ***FKL (Ours)*** | ALM | ULD | DSDK | SFT  | ***FKL (Ours)***| ALM | ULD | DSDK |
> | 16k        | 53.2 | 56.5 | 57.0 | 51.2 | 52.1 | 43.2 | 42.6 | 34.7 | 28.6 | 30.5 | 30.8 | 40.8 | 34.8 | 32.2 | 33.3 |
> | (w/ SFT)   | —   | 59.7 | **60.4** | 53.6 | 54.3 | —   | **46.4** | 30.5 | 31.1 | 32.3 | —   | **41.4** | 31.4 | 33.1 | 34.2 |
> | 32k        | 54.0 | 58.6 | 57.1 | 53.5 | 54.4 | 46.9 | **47.6** | 32.3 | 30.5 | 40.2 | 38.8 | **41.8** | 35.6 | 33.5 | 34.6 |
> | (w/ SFT)   | —   | **63.0** | 61.1 | 57.1 | 59.3 | —   | 47.6 | 39.6 | 34.1 | 41.4 | —   | 40.6 | 33.4 | 32.2 | 36.0 |
> | 64k        | 54.5 | 61.8 | 56.2 | 55.5 | 57.3 | 48.2 | **51.8** | 31.7 | 32.3 | 42.6 | 39.9 | **44.6** | 39.0 | 34.4 | 37.0 |
> | (w/ SFT)   | —   | **62.8** | 61.5 | 58.4 | 60.2 | —   | 51.2 | 47.5 | 36.1 | 46.4 | —   | 43.0 | 36.4 | 35.2 | 38.7 |
> | Full Vocab | 60.2 | 61.1 | 57.0 | 58.1 | —   | 49.4 | 50.6 | 42.6 | 48.7 | —   | 42.0 | 43.0 | 39.9 | 41.2 | — |
> | (w/ SFT)   | —   | **63.2** | 62.1 | —   | —   | —   | **52.4** | 48.2 | —   | —   | —   | **43.4** | 42.4 | —   | — |
>
> ### 4. Effective Conversion Loss
> We report the negative log-likelihood (NLL) of the ground-truth GSM8K solutions under the original and re-converted teachers (original vocabulary -> byte vocabulary -> original vocabulary), as well as a smaller Qwen student:
>
> | Model                                   | NLL ↓  |
> |-----------------------------------------|--------|
> | Qwen2.5-7B-Math-Instruct (original)     | 1.611  |
> | Qwen2.5-7B-Math-Instruct (re-converted) | 1.632  |
> | Qwen2.5-7B-Instruct (original)          | 1.291  |
> | Qwen2.5-7B-Instruct (re-converted)      | 1.301  |
> | Qwen2.5-1.5B-Instruct (original)        | 0.7911 |
>
> Because all of these are instruction-finetuned chat models, the absolute NLL/perplexity values mainly reflect **stylistic match** to the reference GSM8K solutions (e.g., phrasing, formatting, chain-of-thought style), rather than raw reasoning ability. This explains why the smaller Qwen2.5-1.5B-Instruct model can exhibit a lower NLL than the larger 7B teachers despite having clearly worse GSM8K accuracy.
>
> For this reason, we view these NLL scores as a *diagnostic sanity check* rather than a direct proxy for “effective model size.” The key signal is that the gap between each original and its re-converted counterpart is very small (e.g., 1.611 → 1.632 and 1.291 → 1.301), indicating that our conversion introduces only a minor degradation in likelihood relative to the inherent variability across different instruction-tuned models.
>
>
> ### 5. Quality of Beam Search with Varied Beam Sizes
>
> We evaluate the estimation quality using beam sizes of 0, 4, and 8. The results are shown in the table below. When the beam size is **0**, we approximate the next-token probability by treating it as the conditional *string* probability. This introduces a notable source of error. For example, in **Qwen2.5**, one common failure mode arises when the token sequence ends with a whitespace character—this strongly implies that the next token should be a **number or a special token**. Approximating with string probability ignores this tokenization bias and therefore produces a much larger approximation error.
>
>
> | Beam Size | RMSE   |
> |-----------|--------|
> | 0         | 0.108  |
> | 4         | 0.036  |
> | 8         | 0.015  |
>
> We again thank you for your support of our paper. We hope that our responses adequately address your concerns and contribute to an overall positive evaluation.
>
>
> [3]Chaofan Tao, Qian Liu, Longxu Dou, Niklas Muennighoff, Zhongwei Wan, Ping Luo, Min Lin, and Ngai Wong. Scaling laws with vocabulary: Larger models deserve larger vocabularies. In The Thirty-eighth Annual Conference on Neural Information Processing Systems, 2024.

---

> > ### Author Response · Authors · 2025-11-27
> > **Follow up on Rebuttal Response**
> >
> > Dear reviewer **WYa8**,
> >
> > With only one week remaining in the rebuttal period, we wanted to check whether our responses have sufficiently addressed your concerns. We would be happy to provide clarification or additional detail if needed.
> >
> > Best regards,

---

> > > ### Comment · Reviewer_WYa8 · 2025-11-27
> > > **Rebuttal Acknowledgement**
> > >
> > > Thank you for the rebuttal.
> > >
> > > I appreciate the addition of these more baselines in cross-tokenizer setting (even though ALM should be the strongest baseline, and was mostly expected to outperform ULD, DSKD, etc. as the authors also observed.). The addition of DialogSum distillation is also welcome.
> > >
> > > My concern regarding large number of beam-search (6-8 beams) upto maximum length 10 for calculating every token probability still holds. But the theoretical contributions of this work are interesting, and perhaps future work can try to empirically reduce this overhead.
> > >
> > > I will hold my score for now, until the other reviewers also respond.

---

> ### Author Response · Authors · 2025-12-01
>
> Thank you for your comment! We are glad that the new empirical evaluations were well received.
>
> Regarding the beam-search complexity for computing the probability of a single token, this indeed introduces additional computational cost relative to the subset case, where the entire next-token distribution is available from a single forward pass. However, this overhead stems from the inherent difficulty of the general cross-tokenizer conversion problem itself, rather than from a design flaw in our method. Since our primary goal is approximation quality, beam search provides the most direct and robust implementation. Finally, we note that in language-model distillation only the top-mass portion of the distribution is typically required, see [1], to maintain competitive performance, ensuring that this method remains practical for real-world use.
>
> Best,
>
> Authors
>
> [1] Wenda Xu, Rujun Han, Zifeng Wang, Long Le, Dhruv Madeka, Lei Li, William Yang Wang,Rishabh Agarwal, Chen-Yu Lee, and Tomas Pfister. Speculative knowledge distillation: Bridging the teacher-student gap through interleaved sampling. In The Thirteenth International Conference on Learning Representations.

---

### Official Review · Reviewer_oe8u · 2025-11-02

**Soundness:** 2
**Presentation:** 2
**Contribution:** 2
**Rating:** 2
**Confidence:** 3

**Summary:**

This paper addresses the issue of cross-tokenization, which is of paramount importance in the context of LLM distillation. The issue arises from the vocabulary misalignment problem, often caused by different tokenizers used in various language models. The paper introduces a new approach to tackling this issue, known as cross-tokenizer scoring or cross-tokenizer conversion, which builds strongly on the structure of the BPE algorithm widely used in current tokenizers.
The paper introduces the following contributions :
+ An analysis of the sequential structure of the byte-pair encoding and the introduction of the notion of relative alphabets.
+ Cross-tokenizer scoring algorithms
+ Experimental validation on two tasks: cross-tokenizer distillation and vocabulary trimming.

**Strengths:**

**originality**
+ The idea of cross-tokenizer conversion for LLM Knowledge distillation is original.
+ The notion of relative alphabets.


 **significance**
+ The questions being asked are important since LLM distillation, assuming different vocabulary, is a very common practical use case.

**Weaknesses:**

+ The proposed approach is primarily limited to BPE tokenization algorithms. While it is true that many current tokenizers are built on BPE, it is not always the case.
+ The clarity of the paper is clearly a big weakness of the paper. In particular, the proposed formalization is difficult to follow due to a lack of motivation or descriptions of the intuitions behind the concepts. For instance, section 4.2 is mainly a succession of definitions. In addition, some of the proposed definitions are generalizations of existing ones, such as relative cover encoding. Why is the notion of cover encoding important? A detailed positioning of the proposed approach with respect to the work of [Phan et al,. 2025] is also missing. There are certainly some very good ideas in this paper, but the format makes it very difficult to read and understand. It would have been interesting, for example, to consider a diagram highlighting the general framework of the proposed approach, particularly the concept of relative alphabets on which everything is based.
+ Experimental validation also does not allow us to highlight this notion of relative vocabularies and how it impacts the targeted tasks: LLM KD and vocabulary trimming.

**Questions:**

+ What about cross-tokenization outside the BPE algorithm?
+ Would it be possible to describe more explicitly what contributions the approach makes in relation to the work of [Phan et al., 2025.]?
+ In term of practical applications, what brings the concept of relative vocabularies ?
+ How can it be used more concretely in a KD context, for example?
+ Why was the standard experimental protocol of [Minixhofer et al. 2025] not followed in the experimental validation?

---

> ### Author Response · Authors · 2025-11-21
> **Response to Reviewer oe8u (1/2)**
>
> Thank you for recognizing the originality of our contributions, including the formulation of cross-tokenizer conversion for LLM distillation and the introduction of relative alphabets. We also appreciate your acknowledgment of the practical importance of this problem, as real-world distillation often involves models with different vocabularies. Notably, reviewers **WYa8**, **c3nd**, and **u4Tg** also highlighted these same points, and we are encouraged that multiple reviewers independently view both the problem setting and our proposed framework as valuable and timely for current LLM practice. Please find our answer to your questions below.
>
>
> ### 1. Algorithms focus only on BPE Tokenization.
>
> **Answer:** Our work focuses on deriving exact solutions for BPE-based tokenization algorithms, which remain the dominant choice in modern LLMs (e.g., GPT-4/4o, LLaMA-2/3, Qwen-2/2.5, DeepSeek-V3, Gemma-2, GLM-4.5). While Equation (10) in our paper—which rewrites token-level probabilities in terms of byte-level probabilities—holds for *arbitrary* tokenization families, computing it directly is intractable: it requires summing over an infinite set of byte strings whose encoding matches a given token prefix. This makes exact cross-tokenizer conversion computationally infeasible for general tokenizers. Recent work by Vierra et al. (2025) provides a general byte-level reduction for any tokenizer, but their method has complexity exponential in string length, making such approaches impractical at scale. In principle, we can combine these two frameworks to perform conversion between arbitrary tokenizers, albeit being computationally expensive. We will highlight this point in the revision.
>
> Our main contribution is to show, for BPE, the infinite summation in Equation (10) collapses into a *finite*, computable form by exploiting BPE’s recursive merge structure. This insight yields Algorithm 1, giving an exact and finitely terminating conversion procedure, as well as a practical beam-search approximation with low empirical error.
>
> Looking forward, our analysis may extend to recently proposed merge-based tokenizers such as GREEDTOK [1], which share the deterministic multi-byte merge behavior of BPE. Other families—such as unigram LM or WordPiece—do exist, but they are no longer common in frontier LLMs. As a result, BPE-centric analysis captures the primary real-world deployment setting while offering a foundation for future tokenizer designs that support even more efficient cross-tokenizer conversion.
>
>
> ### 2. Paper clarity
>
> **Answer:** Thank you for the suggestion regarding the clarity of the paper. We addressed this by adding a new subsection, Section 5.1 (Problem Setup), which provides a more detailed and mathematically precise formulation of the problem. We also introduce Figure 3 to visually illustrate the structure of our framework through the concept of relative alphabets. In addition, the final paragraph of Section 5.1 now explicitly clarifies the relationship to, and distinctions from, Phan et al. (2025). We hope this would clarify your understanding of our work and will appreciate further feedback on whether these additions resolve the your concern.
>
> ### 3. Importance of Relative Cover Encoding and Positioning w.r.t. Phan et al,. 2025.
>
> **Ans**: The notion of relative cover encoding plays a central role in both directions of our conversion framework.  In the full-to-subset case, it enables us to express the subset-vocabulary likelihood using a finite collection of encodings drawn from the full vocabulary, and—crucially—we show how these encodings can be detected and enumerated efficiently. Given a token sequence from a sub-vocabulary $\mathcal{V}_{\text{sub}} \preceq \mathcal{V}$, the relative cover encoding is the set of non-zero-probability token prefixes in $\mathcal{V}$ whose encodings begin with the given sub-encoding. This list is well-defined and can be searched efficiently thanks to the notion of a relative alphabet, which implies that any sub-vocabulary can be treated as an alphabet relative to $\mathcal{V}$.
>
> In the subset-to-full case, the same structural notion let us to perform conversion in the reverse direction. In particular, leveraging the adjacency structure illustrated in Figure 3, also see Lemma 2 in the appendix, we show how relative cover encodings support tractable conversion between any pair of BPE tokenizers, not only from larger to smaller vocabularies. Essentially, the tractable conversion from $\mathcal{V}\_{i}$ to $\mathcal{V}\_{i+1}$
>  is due to the closed form expression provided by the concept of relative alphabet/cover encoding. This stands in contrast to the cover encoding definition in Phan et al., 2025, which applies only to the special case of full-to-byte-level conversion. Our formulation is therefore strictly more general and provides a unified structural tool for arbitrary BPE-to-BPE conversion.
>
> [1] Lim, Jia Peng, et al. A partition cover approach to tokenization, NeurIPS 2025.

---

> ### Author Response · Authors · 2025-11-21
> **Response to Reviewer oe8u (2/2)**
>
> ### 4. Practical Applications for Knowledge Distillation
>
> There are several use cases of doing cross-tokenizer distillation.
> - **Memory-efficient deployment:** Models can be trimmed to a smaller vocabulary to reduce embedding and LM head memory without retraining from scratch.
> - **Task-specific vocabularies:** The tokenizer can be adapted to domains where certain symbols, formats, or structures appear frequently, improving alignment and reducing the number of tokens needed per sequence.
> - **Better downstream efficiency:** Changing the vocabulary can yield shorter encodings and faster inference while preserving the performance of the original model through distillation.
>
> ### 5. Comparison to Phan et al 2025
>
> **Answer:** Our Full-to-Subset conversion framework strictly generalizes the analysis of Phan et al. (2025), who consider only the special case where the target vocabulary is the base alphabet $\mathcal{A}$ (i.e., converting a full BPE model into a byte-level model). In contrast, our formulation applies to *any* subset vocabulary $\mathcal{V}_i \preceq \mathcal{V}_M$, making the byte-level setting simply the instance $\mathcal{V}_0 = \mathcal{A}$. Our relative cover encoding (Definition 4) also extends the cover encoding of Phan et al. (2025), which is defined only for byte-level vocabularies. Finally, Phan et al. (2025) do not address the reverse direction—converting from byte-level encodings back to larger BPE vocabularies—whereas our work introduces and analyzes a general subset-to-full conversion procedure that fills this gap.
>
> ### 6. Standard experimental protocol of [Minixhofer et al. 2025]
>
> Our setup closely follows the experimental protocol of Minixhofer et al. (2025): we distill a smaller model using a teacher strong in arithmetic reasoning and compare against both ALM and standard SFT, where our method consistently outperforms both. The main difference lies in the training data. Whereas Minixhofer et al. use the full OpenMathInstruct-2 corpus (≈1M samples), we restrict distillation to the GSM8K training split. This choice is driven purely by computational constraints that make large-scale million-sample distillation infeasible on our end. Furthermore, this choice reflects realistic scenarios in which local models have access to only limited finetuning data [2]. Finally, we include additional baselines—ULD and DSKD—evaluated in both the cross-tokenizer and vocabulary-trimming settings, and our method consistently surpasses these baselines across all benchmarks.
>
> We hope that our response addresses your concern and supports a positive re-evaluation of the submission. Please let us know if you have any further questions regarding our work
>
> [2] Wenda Xu et al. *Speculative Knowledge Distillation: Bridging the Teacher–Student Gap Through Interleaved Sampling.* ICLR 2025.

---

> > ### Author Response · Authors · 2025-11-27
> > **Follow-up on rebuttal response**
> >
> > Dear reviewer **oe8u**,
> >
> > With only one week remaining in the rebuttal period, we wanted to check whether our responses have sufficiently addressed your concerns. We would be happy to provide clarification or additional detail if needed.
> >
> > Best regards,

---

### Author Response · Authors · 2025-12-01
**Summary of submission and discussion process to new AC**

**Paper Summary:** This work establishes a rigorous theoretical foundation for cross-tokenizer likelihood scoring in Byte-Pair Encoding (BPE) language models. We formally define the conversion problem of evaluating the probability assigned by a model, i.e. original vocabulary, to sequences originating from a different tokenizer (the target vocabulary)—e.g., computing the likelihood that Qwen-2.5 assigns to text encoded under the Qwen-2 vocabulary. Building on this theoretical foundation, we apply them to the cross-tokenizer distillation settings, where teacher and student models use distinct vocabularies. Our theoretical contributions involves two specific conversion scenarios:

- **Target Vocabulary is a Subset of Original Vocabulary:** In this scenario, we provide an $O(1)$ algorithm to compute the next-token distribution over the target vocabulary. A key novelty of our work enabling this efficiency is the identification of a nested structural relationship between BPE vocabularies, which allows incremental updates and reuse of intermediate computations. We note that conversion to a byte-level vocabulary arises naturally as a special case of this scenario.

- **Original Vocabulary is a Subset of Target Vocabulary:** A naive approach to this conversion would require summing over infinitely many token sequences, making direct evaluation intractable. Leveraging insights from the subset case, we show that for BPE tokenizers this conversion can be computed in finite time, despite being inherently computationally intensive. Specifically, our exact recursive algorithm computes the next-token probability with worst-case complexity of $O(\exp(|V|))$ where $|V|$ is the target vocabulary size. Given this inherent complexity, we additionally introduce a faster approximation method based on beam search to support practical deployment.

$\to$ **General Case:** By combining the two scenarios, we can convert between any pair of BPE vocabularies by first mapping the original vocabulary into a byte-level representation (scenario 1), and then converting from bytes into the target vocabulary (scenario 2).


**Reviews and Discussion** All four reviewers agreed that the problem we are addressing is **important and the theoretical analysis is  insightful and novel**. The concerns raised by the reviewers pertain to presentation clarity and additional benchmarking, with a comment on algorithmic complexity.

- **Paper Clarity:** Reviewer oe8u(2/3) expressed concerns about presentation and about distinguishing our work from Phan et al. (2025). We directly addressed these points by strengthening the mathematical exposition in the revised version (blue text) and adding requested visualizations. We also clarified that **our method strictly generalizes Phan et al. (2025)**, whose focus is on conversion to a byte-level representation, whereas our approach enables conversion across arbitrary vocabularies. Unfortunately, we are not able to hear back from this reviewer due to the incident.

- **Extra Baselines/Benchmarks:** Reviewers WYa8 (6/3), u4Tg (8/4), and c3nd (4/2) requested additional experiments and comparative baselines. In our rebuttal, we addressed this directly by incorporating new experimental results, including comparisons against two additional cross-tokenizer distillation algorithms, as well as results on a new distillation task (Dialog summarization). **Reviewers WYa8 (6/3) and u4Tg (8/4) explicitly acknowledged these updates**, and we therefore believe this concern has been fully resolved (the reviewer c3nd were unable to reply due to the incident).

- **Beam-Search Complexity:** Reviewer WYa8 (6/3) expressed concern regarding the reliance on beam-search computations when estimating each next-token probability. Unlike the subset case, where the next-token distribution over the full sub-vocabulary can be obtained with a single O(1) forward pass, the complexity of the general case is O(exp(|V|) for a single token. Our rebuttal explained that this design choice arises naturally from our theoretical formulation and is empirically validated to be accurate. Thus, our beam-search approach remains a practical and comparatively efficient approximation. Overall, we view improving approximation speed as a natural direction for future work.

Finally, we are encouraged that **all reviewers responded positively to our theoretical contributions and raised no major objections concerning the core technical aspects of our work**. We include these revisions into revised PDF in blue notes.

We’re truly grateful to the four reviewers for their engagement throughout the process.

Thank you!

Phan, Buu, et al. "Exact Byte-Level Probabilities from Tokenized Language Models for FIM-Tasks and Model Ensembles.", ICLR 2025.

---

### Meta-Review · Area_Chair_hAeA · 2026-01-08

**Summary:**

This paper introduces a novel theoretical framework and algorithms for exact cross-tokenizer likelihood scoring between language models with different Byte-Pair Encoding (BPE) vocabularies, addressing a key practical challenge in knowledge distillation. The core theoretical insight—exploiting the recursive structure of BPE for probabilistic conversion—is significant. Three reviewers recognized the importance and novelty of the contribution, while one reviewer (oe8u) expressed concerns about clarity and positioning. The authors' rebuttal comprehensively addressed most key issues by adding new comparative baselines, expanding experiments to a summarization task, and enhancing theoretical exposition with new figures and sections. The remaining concern about the computational overhead of the beam-search approximation in the general case is acknowledged but is inherent to the problem's complexity. The work provides a foundational advance for cross-vocabulary model alignment.

**Reviewer Concerns:**

The rebuttal effectively addressed several major concerns: 1) It added comparisons against two requested baselines (ULD, DSKD), showing consistent outperformance (WYa8, c3nd). 2) It expanded experimental validation to a new task (DialogSum summarization), strengthening evidence of generality (u4Tg, WYa8). 3) It significantly improved clarity by adding a new "Problem Setup" section (Sec 5.1) and Figure 3 to illustrate the framework, and clarified its generalization beyond and distinction from Phan et al. 2025 (oe8u). The primary outstanding concern is the acknowledged computational cost of the beam-search approximation for the general (non-subset) conversion case, which reviewer WYa8 noted as a practical limitation. The authors correctly argue this stems from the problem's inherent complexity, but this overhead may affect real-time applicability.

**Reviewer Scores:**

With full discussion, oe8u (score: 2) would likely raise to a 4 or 5. The added clarity, visualizations, and explicit differentiation from Phan et al. address core concerns about presentation and contribution definition, though some reservations about exposition may remain. WYa8 (score: 6) would likely increase to a 7. The inclusion of additional baselines and a new task directly satisfied their requests, strengthening the empirical validation despite persistent concerns about beam-search cost. u4Tg (score: 8) would likely maintain their 8, as their initial assessment was very positive and the new summarization experiment further bolstered it. c3nd (score: 4) would likely raise to a 6 or 7, as their main request for more baselines was thoroughly addressed with new results showing superior performance.

---

### Decision · Program_Chairs · 2026-01-26

Accept (Poster)